# The viral context instructs the redundancy of costimulatory pathways in driving CD8+ T cell expansion

Suzanne PM Welten[1], Anke Redeker[1], Kees LMC Franken[1], Jennifer D Oduro[2], Ferry Ossendorp[1], Luka Čičin-Šain[2,3], Cornelis JM Melief[1,4], Peter Aichele[5], Ramon Arens[1]*

[1]Department of Immunohematology and Blood Transfusion, Leiden University Medical Center, Leiden, Netherlands; [2]Department for Vaccinology/Immune Aging and Chronic Infection, Helmholtz-Zentrum für Infektionsforschung GmbH, Braunschweig, Germany; [3]Department for Virology, Medical School Hannover, Hannover, Germany; [4]ISA Pharmaceuticals, Leiden, Netherlands; [5]Department of Medical Microbiology and Hygiene, Institute of Immunology, University of Freiburg, Freiburg, Germany

**Abstract** Signals delivered by costimulatory molecules are implicated in driving T cell expansion. The requirements for these signals, however, vary from dispensable to essential in different infections. We examined the underlying mechanisms of this differential T cell costimulation dependence and found that the viral context determined the dependence on CD28/B7-mediated costimulation for expansion of naive and memory CD8+ T cells, indicating that the requirement for costimulatory signals is not imprinted. Notably, related to the high-level costimulatory molecule expression induced by lymphocytic choriomeningitis virus (LCMV), CD28/B7-mediated costimulation was dispensable for accumulation of LCMV-specific CD8+ T cells because of redundancy with the costimulatory pathways induced by TNF receptor family members (i.e., CD27, OX40, and 4-1BB). Type I IFN signaling in viral-specific CD8+ T cells is slightly redundant with costimulatory signals. These results highlight that pathogen-specific conditions differentially and uniquely dictate the utilization of costimulatory pathways allowing shaping of effector and memory antigen-specific CD8+ T cell responses.

*For correspondence: R.Arens@lumc.nl

Competing interests: The authors declare that no competing interests exist.

## Introduction

CD8+ T cells are critical for elimination of various intracellular pathogens. By incorporating differences in TCR signal strength and duration (signal 1), the spatiotemporal availability of costimulatory molecules (signal 2) and defined cytokines in the inflammatory environment (signal 3), CD8+ T cells are differentially programmed for expansion and effector cell formation resulting in considerable plasticity of the response (*Williams and Bevan, 2007*; *Arens and Schoenberger, 2010*).

Costimulatory molecules augment TCR triggering but also qualitatively contribute to achieve optimal T cell expansion and differentiation (*Croft, 2003*). CD28 is considered as the most prominent costimulatory receptor for T cells, but signals provided by members of the TNF receptor (TNFR) super family such as CD27, OX40 (CD134) and 4-1BB (CD137) are known to provide crucial signals as well. T cell responses seem to be differentially and contextually dependent on costimulatory interactions but the underlying mechanisms are unknown (*DeBenedette et al., 1999*; *Welten et al., 2013a*; *Wortzman et al., 2013*). For example, the pathogen-specific CD8+ T cell response during vesicular

**eLife digest** When the immune system detects a virus in the body it mounts a response to eliminate it. Immune cells called CD8[+] T cells detect fragments of virus proteins that are presented on the surface of other immune cells. The CD8[+] T cells then rapidly divide to form populations that roam the body to kill cells that are infected with the virus. Afterwards, some of the CD8[+] T cells become 'memory T cells', which allow the immune system to respond more rapidly if the virus returns. This means that a subsequent infection of the same virus is usually stopped before it can become severe enough for an individual to feel unwell.

Vaccines take advantage of the activities of CD8[+] T cells to enable a person to become 'immune' to a virus without having to experience the disease. Vaccines contain dead or weakened viruses that can't spread in the body, but are able to activate the CD8[+] T cells. However, a vaccine may not be as effective in activating the T cells as the live virus, perhaps because it fails to trigger the production of other molecules in the host that promote T cell activation. There are many of these 'co-stimulatory molecules' in the body, but it is not clear exactly how they work.

Now, Welten et al. show that the role of co-stimulatory molecules in the activation of CD8[+] T cells depends on the type of virus and how it affects cells. Mice that were genetically engineered to lack two co-stimulatory molecules called CD80 and CD86 failed to accumulate active CD8[+] T cells in response to infection with a herpes-like virus. However, if these mice were infected with a different virus called LCMV—which causes swelling of the brain and spinal cord—they produced many active CD8[+] T cells to fight the infection.

Welten et al. found that other co-stimulatory molecules are able to compensate for the loss of CD80 and CD86 to boost the activation of T cells in response to LCMV, but not the herpes-like virus. Further experiments showed that LCMV triggers a lot more inflammation in infected cells than the other virus. This leads to the production of many different types of co-stimulatory molecules, which ensures that if one fails to boost the activation of CD8[+] T cells, another molecule can do so instead. Better understanding of how these co-stimulatory molecules work could help scientists to develop more effective vaccines in future.

stomatitis virus and vaccinia virus (VV) infection is highly driven by interactions between CD28 and the B7 molecules B7.1 (CD80) and B7.2 (CD86) (*Sigal et al., 1998*; *Bertram et al., 2002*; *Fuse et al., 2008*), while in lymphocytic choriomeningitis virus (LCMV) infection the viral-specific CD8[+] T cells seem to bypass the requirements of the CD28/B7 costimulatory pathway for primary effector T cell expansion (*Shahinian et al., 1993*; *Kundig et al., 1996*; *Andreasen et al., 2000*; *Grujic et al., 2010*; *Eberlein et al., 2012*). Even within a single infection distinct requirements for costimulatory signals can be observed. In mouse cytomegalovirus (MCMV), the classical (non-inflationary) CD8[+] T cell responses are more dependent on the CD28/B7 costimulatory pathway than the so-called inflationary CD8[+] T cells, which gradually accumulate at high frequencies in time (*Arens et al., 2011b*; *O'Hara et al., 2012*).

Here we examined the mechanisms of CD8[+] T cell costimulation dependency. We found that the pathogen-induced environment and not the characteristics of the viral epitopes determined the requirements of naive and of memory CD8[+] T cells for CD28/B7-mediated costimulation. Remarkably, related to the induction of high costimulatory ligand expression, LCMV-specific CD8[+] T cell expansion can operate in a CD28/B7 independent fashion because of redundancy with the costimulatory members of the TNFR superfamily. Furthermore, direct type I IFN signaling in viral-specific CD8[+] T cells is slightly redundant with CD28/B7 and CD27/CD70-mediated costimulation. These findings demonstrate that the inflammatory environment dictates the characteristics of CD8[+] T cell responses by allowing a differential utilization of stimulatory pathways.

## Results

### Differential requirements for CD28/B7-mediated costimulation in driving CD8[+] T cell expansion

Effector CD8[+] T cell formation during LCMV infection seems not to be driven by the main costimulatory CD28/B7 pathway because wild-type (WT) mice and mice deficient in both B7.1 and

B7.2 ($Cd80/86^{-/-}$) mount similar antigen-specific responses in magnitude, and this phenomenon is apparent after both high and low viral inoculum dosages (*Figure 1A*). In contrast, during infection with VV or *Listeria monocytogenes* (LM), antigen-specific CD8+ T cell responses are highly reduced in the absence of B7-mediated costimulation (*Figure 1B,C*). CD8+ T cell responses against MCMV are dependent on B7-mediated costimulation as well, ranging from ∼sevenfold diminished responses in case of the non-inflationary M45 and M57-specific to ∼2.5-fold in case of the inflationary m139 and M38-specific responses (*Figure 1D*). Effector cell differentiation of virus-specific CD8+ T cells, indicated by the downregulation of CD62L and upregulation of CD44, also required B7-mediated costimulation in MCMV but not in LCMV infection (*Figure 1—figure supplement 1*). Thus, in various infections but not during LCMV infection the CD28/B7 costimulatory pathway is highly critical in driving T cell expansion.

Next, we examined if additional triggering of the CD28/B7 costimulatory pathway is able to differentially modulate effector T cell formation. Therefore, the co-inhibitory receptor CTLA-4 that binds to B7.1 and B7.2 was blocked with antibodies during infection, which increases the availability of

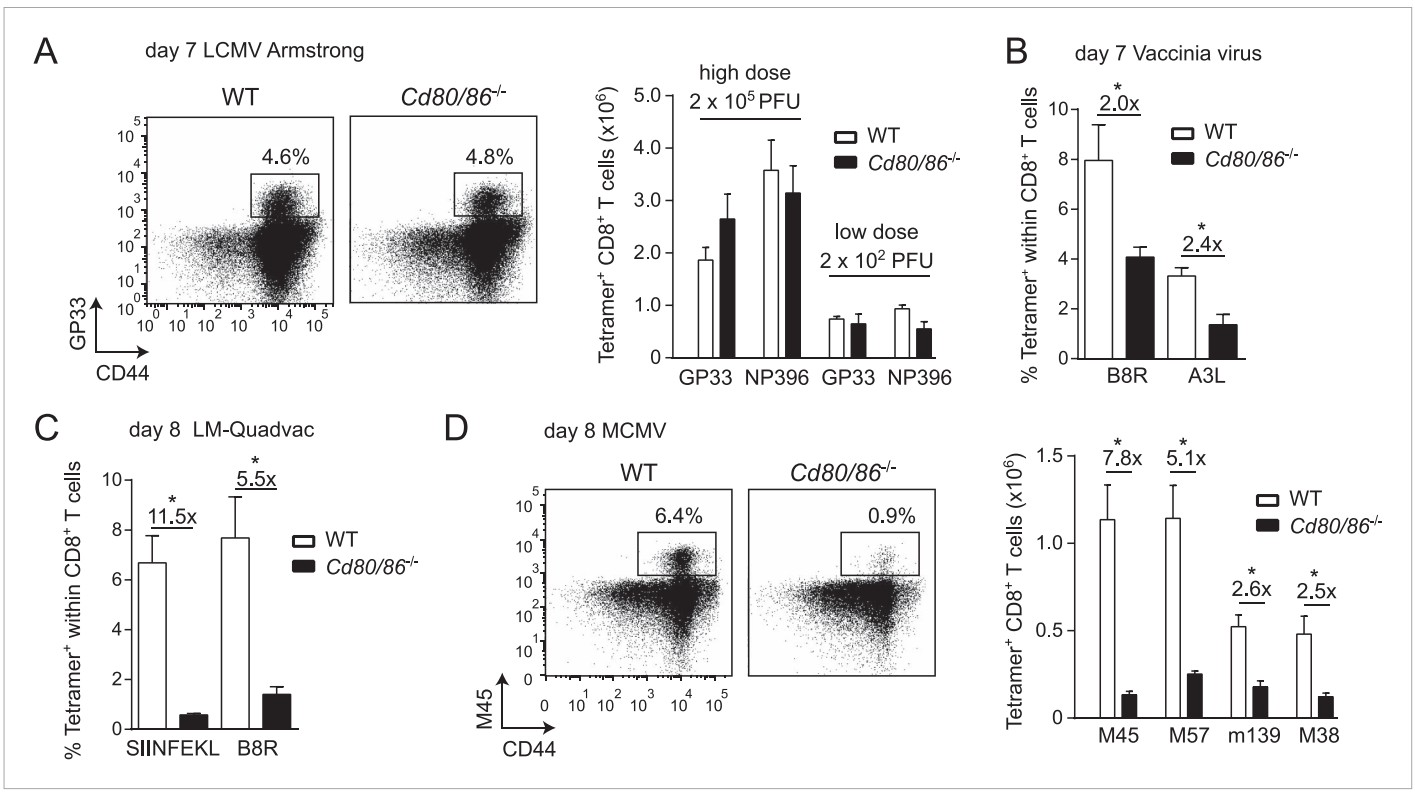

**Figure 1**. Differential requirements for CD28/B7-mediated costimulation in driving pathogen-specific CD8+ T cell expansion. (**A**) Wild-type (WT) and $Cd80/86^{-/-}$ mice were infected with 2 × 10² (low dose) or 2 × 10⁵ (high dose) PFU LCMV-Armstrong. The lymphocytic choriomeningitis virus (LCMV)-specific CD8+ T cell response in the spleen was determined 7 days post-infection. Representative flow cytometric plots show CD3+/CD8+ cells that were stained with CD44 antibodies and MHC class I tetramers (high dose infection). Percentages indicate tetramer+ cells within the CD8+ T cell population. Bar graph shows total number of splenic LCMV-specific CD8+ T cells. (**B**) Mice were infected with 2 × 10⁵ PFU vaccinia virus (VV) WR and the percentage of tetramer+ cells within the CD8+ T cell population was determined in the blood 7 days post-infection. (**C**) The percentage of tetramer+ cells within the CD8+ T cell population was determined in the blood 7 days post-infection with 1 × 10⁶ CFU LM-Quadvac. (**D**) Flow cytometric plots show a representative M45-specific tetramer staining of cells from WT and $Cd80/86^{-/-}$ mice at day 8 post-infection with 1 × 10⁴ PFU mouse cytomegalovirus (MCMV). Cells are gated on CD3+/CD8+ and the percentages indicate tetramer+ cells within the CD3+/CD8+ T cell population. Bar graph indicates the total number of splenic MCMV-specific CD8+ T cells. Data in bar graphs are expressed as mean + standard error of the mean (SEM) (n = 5–12 mice per group) of at least two independent experiments. Fold difference and significance (*p < 0.05) is indicated.

The following figure supplement is available for figure 1:

**Figure supplement 1**. Costimulatory signals program effector cell differentiation of MCMV-specific but not of LCMV-specific CD8+ T cells.

the B7 molecules to stimulate CD28. Remarkably, CTLA-4 blockade during LCMV infection had no effect on T cell expansion, indicating that LCMV-specific CD8+ T cells are rather indifferent to enhanced B7-mediated signals (*Figure 2A,B*). However, CTLA-4 blockade during MCMV infection augmented MCMV-specific CD8+ T cell responses ~threefold in a B7-dependent manner (*Figure 2C, D*). Thus, additional triggering of the CD28/B7 pathway is beneficial in settings in which T cell expansion is dependent on this pathway, while the enhancement of CD28/B7-mediated costimulation had no effect in conditions in which the B7 costimulatory molecules are not essential for initial T cell expansion.

### The context of viral epitope expression determines the requirement for CD28/B7-mediated costimulation in driving T cell expansion

To determine whether the characteristics of LCMV-specific epitopes define the B7-independent activation of CD8+ T cell responses, we analyzed the response to the immunodominant epitope GP$_{33-41}$ of LCMV (GP33) in the context of different pathogen infections. Therefore, recombinant MCMVs were generated in which the GP33 epitope was expressed within the immediate early 2 (IE2) protein (MCMV-IE2-GP33) or the M45 protein (MCMV-M45-GP33). The in vitro replication kinetics of MCMV-IE2-GP33 and MCMV-M45-GP33 were similar as WT virus (*Figure 3—figure supplement 1A*). Correspondingly, in vivo infection with MCMV-IE2-GP33 induced a GP33-specific response with inflationary characteristics, as specified by a gradual increasing GP33-specific CD8+ T cell response in time with an effector memory phenotype (*Figure 3—figure supplement 1B,C*). As determined by intracellular IFN-γ staining after restimulation (*Figure 3A,B*) or direct staining with MHC class I tetramers (data not shown), the GP33-specific CD8+ T cell response elicited by both MCMV-IE2-GP33 and MCMV-M45-GP33 was dependent on B7-mediated costimulation, albeit to a higher degree when the GP33 epitope was inserted within the M45 protein. Infection with an MCMV containing the model epitope OVA$_{257-264}$ (SIINFEKL) inserted in the M45 protein (MCMV-M45-SIINFEKL) resulted also in an antigen-specific T cell response that depended on B7-mediated costimulation (*Figure 3C*), indicating that non-viral epitopes elicit similar costimulation dependent responses. Furthermore, LM expressing the LCMV GP33 epitope (LM-GP33) induced GP33-specific CD8+ T cell responses that were highly dependent on B7-mediated costimulation (*Figure 3D*). Also, upon vaccination with a synthetic long peptide (SLP) containing the GP33 epitope, *Cd80/86$^{-/-}$* mice mounted a defective GP33-specific

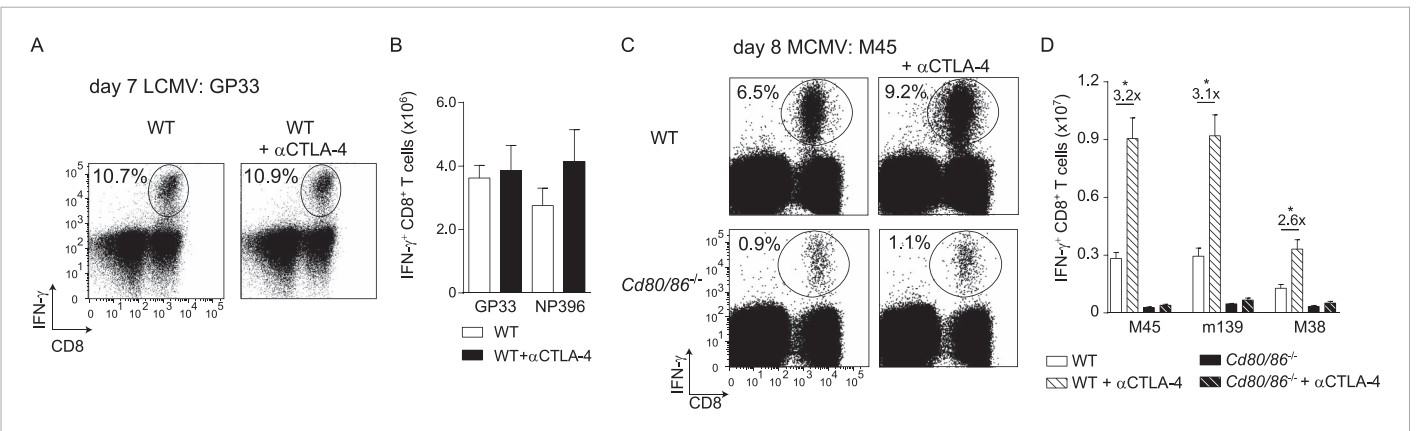

**Figure 2**. CTLA-4 blockade impacts B7-driven CD8+ T cell responses. (**A**) CTLA-4 blocking antibodies were administrated during infection with $2 \times 10^5$ PFU LCMV Armstrong in WT mice. At day 7 post-infection, the splenic LCMV-specific response was analyzed by intracellular cytokine staining. Representative flow cytometric plots show intracellular IFN-γ vs cell-surface CD8 staining after restimulation with GP$_{33-41}$ peptide. The percentage of IFN-γ+ cells within the CD8+ T cell population is indicated. (**B**) Total numbers of splenic LCMV-specific CD8+ T cells are shown. (**C**) CTLA-4 interactions were abrogated by administration of blocking antibodies in WT and *Cd80/86$^{-/-}$* mice upon infection with $1 \times 10^4$ PFU MCMV, and at day 8 post-infection the virus-specific response was analyzed by intracellular cytokine staining. Representative flow cytometric plots show intracellular IFN-γ vs CD8 staining after restimulation of splenocytes with M45$_{985-993}$ peptide. The percentage of IFN-γ+ cells within the CD8+ T cell population is indicated. (**D**) Total numbers of MCMV-specific CD8+ T cells in the spleen are shown. Data in bar graphs are expressed as mean + SEM (n = 4–5 mice per group) of at least two independent experiments. Fold difference and significance (*p < 0.05) is indicated.

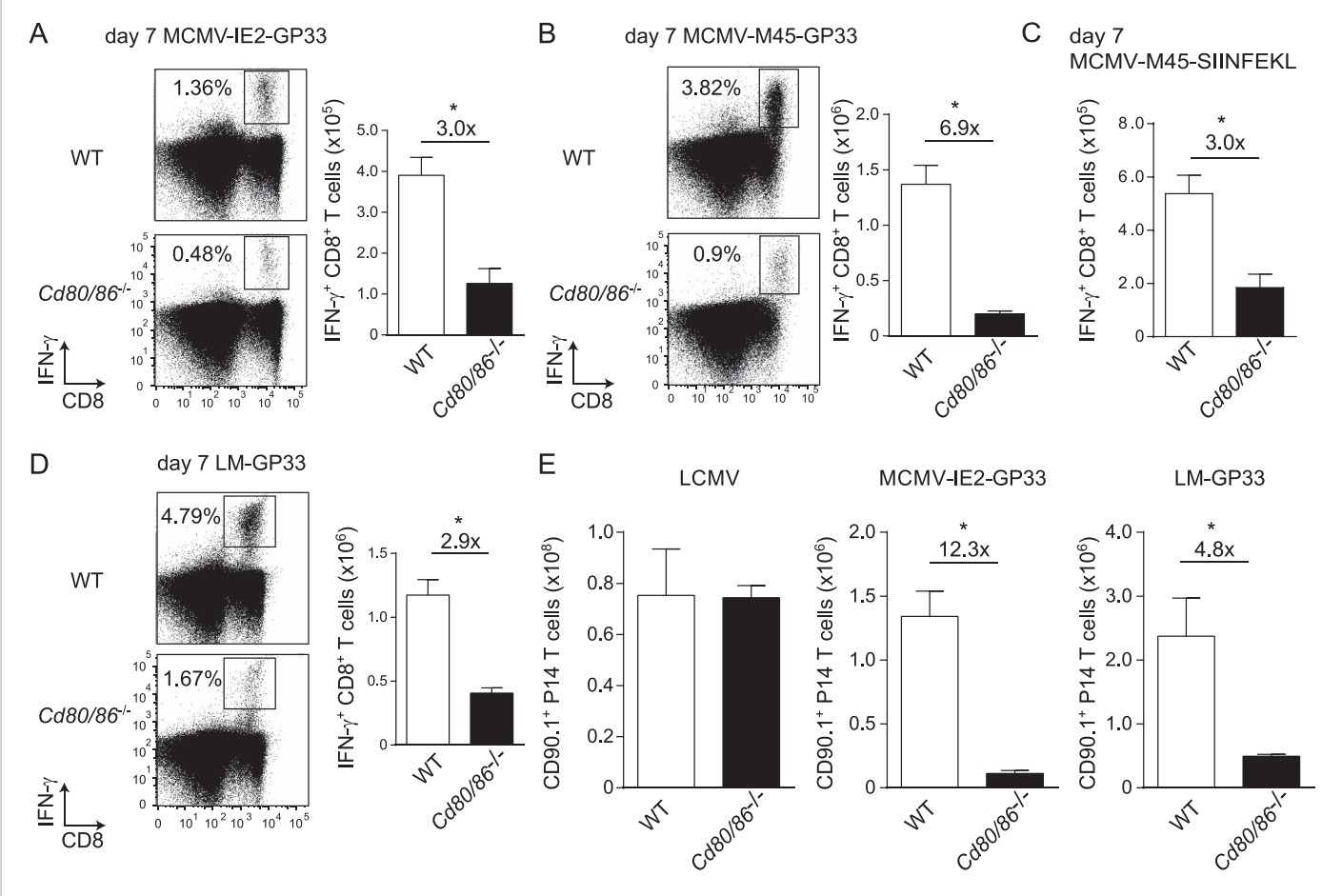

**Figure 3**. The context of viral epitope expression determines the requirements for B7-mediated costimulation in driving antigen-specific CD8+ T cell expansion. (**A**, **B**) WT and *Cd80/86*−/− mice were infected with 1 × 10^5 PFU MCMV-IE2-GP33 or MCMV-M45-GP33, and 8 days post-infection the splenic GP33-specific CD8+ T cell response was determined by intracellular IFN-γ staining. Representative flow cytometric plots are shown and the percentage of IFN-γ+ cells within the CD8+ T cell population is indicated. Graphs indicate the total number of splenic GP33-specific CD8+ T cells. (**C**) The splenic SIINFKEL-specific CD8+ T cell response was determined by intracellular IFN-γ staining at day 8 post-infection with 1 × 10^5 PFU MCMV-M45-SIINFEKL. (**D**) WT and *Cd80/86*−/− mice were infected with 1.5 × 10^3 CFU LM-GP33. At day 7 post-infection the splenic GP33-specific response was analyzed by intracellular IFN-γ staining upon restimulation with GP33 peptide. Representative flow cytometric plots are shown. The percentage of GP33-specific CD8+ T cells within the total CD8+ T cell population is indicated. Bar graph shows the total number of splenic GP33-specific CD8+ T cells. (**E**) 5 × 10^4 CD90.1+ *Ifnar1*+/+ P14 cells were adoptively transferred into WT and *Cd80/86*−/− mice that were subsequently infected with 2 × 10^5 PFU LCMV Armstrong, 1 × 10^5 PFU MCMV-IE2-GP33 or 1.5 × 10^3 CFU LM-GP33. At day 7 (LCMV, LM-GP33) or 8 (MCMV-IE2-GP33) post-infection, the magnitude of the P14 cell response in the spleen was determined. Data in bar graphs are expressed as mean + SEM (n = 3–5 mice per group) of at least two independent experiments. Fold difference and significance (*p < 0.05) is indicated.

The following figure supplements are available for figure 3:

**Figure supplement 1**. Characteristics of recombinant MCMVs.

**Figure supplement 2**. GP33-SLP vaccination is dependent on B7-mediated costimulation.

CD8+ T cell response in comparison with WT mice (**Figure 3—figure supplement 2**). To exclude possible effects related to differences in the TCR repertoire selection, TCR transgenic CD8+ T cells recognizing the LCMV GP33 epitope (referred hereafter as P14 cells) were used in different pathogenic contexts. Similar as observed for the endogenous LCMV-specific CD8+ T cell expansion, B7-mediated costimulation was dispensable for P14 cell expansion in LCMV infection. Importantly, for the expansion of P14 cells in MCMV-IE2-GP33 and LM-GP33 infection, B7-mediated signals were highly required (**Figure 3E**), which corroborates that the inflammatory environment is predominantly

determining the costimulatory requirements. Together, these data indicate that the context of viral epitope expression, rather than the intrinsic nature of the epitope or the antigen-specific CD8[+] T cell population, influences the dependence on B7-mediated signals for T cell expansion.

## The context of viral epitope expression during secondary expansion determines the requirement for CD28/B7-mediated costimulation independent of the priming context

Next, we examined whether the viral context imprints the costimulatory requirements for the lifespan of T cells or if memory T cells undergo secondary expansion independently of the priming context. Therefore, crisscross adoptive transfer experiments were performed with memory GP33-specific CD8[+] T cells generated in different viral environments. First, memory GP33-specific CD8[+] T cells were primed in the context of an LCMV infection and adoptively transferred into WT or Cd80/86[−/−] hosts that were subsequently infected with either LCMV or MCMV-IE2-GP33. Comparing the GP33-specific CD8[+] T cell responses upon antigenic re-challenge revealed a dispensable role for B7-mediated signals for secondary T cell expansion in an LCMV environment, but a strong requirement for these signals in an MCMV context even though these cells were primed in an LCMV environment (*Figure 4A*). Importantly, when memory GP33-specific CD8[+] T cells that depended on B7-mediated signals during priming in MCMV infection were transferred and re-challenged in an LCMV or an MCMV environment, the viral context during secondary expansion again determined the requirement on costimulatory signals (*Figure 4B*). Together, these data indicate that during secondary T cell responses the viral context is dominant and determines the CD28/B7 costimulation dependency of virus-specific CD8[+] T cells independent of the priming context.

## Influence of type I IFN signaling on the requirement of CD28/B7-mediated costimulation

As the context of a viral infection determines the dependence on CD28/B7-mediated costimulation for CD8[+] T cell expansion, we compared the overall composition of inflammatory mediators in LCMV

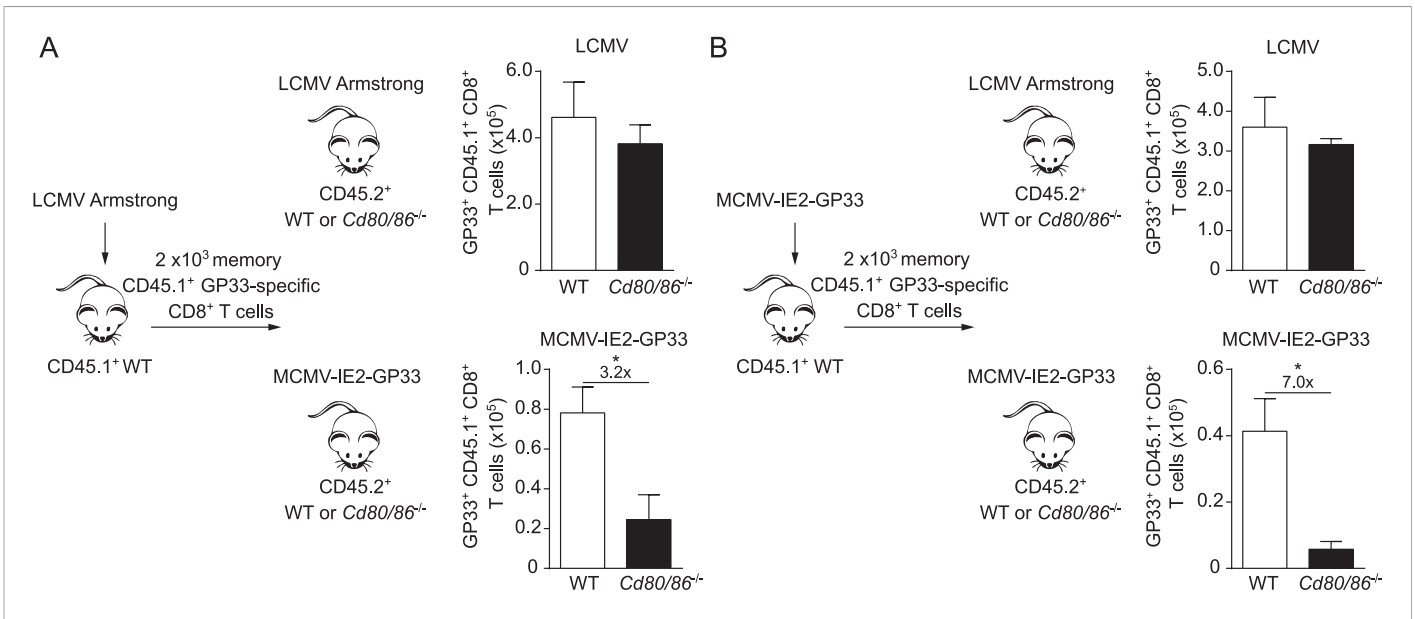

**Figure 4**. The infectious pathogen during antigenic re-challenge determines the requirements for CD28/B7-mediated costimulation for secondary expansion. (**A**) Experimental setup: CD45.1[+] WT mice were infected with $2 \times 10^5$ PFU LCMV Armstrong. After 4 months GP33-specific memory CD8[+] T cells were sorted and $2 \times 10^3$ cells were adoptively transferred into CD45.2[+] WT and Cd80/86[−/−] mice that were subsequently infected with $2 \times 10^5$ PFU LCMV Armstrong or $1 \times 10^5$ PFU MCMV-IE2-GP33. The total number of transferred GP33-specific CD8[+] T cells was determined 6 days post re-challenge. (**B**) Similar experimental setup as described in (**A**), except CD45.1[+] WT mice were infected with $1 \times 10^5$ PFU MCMV-IE2-GP33. Data in bar graphs are expressed as mean + SEM (n = 5 mice per group). Fold difference and significance (*p < 0.05) is indicated.

and MCMV infection. Expression of the inflammation-associated cytokines IL-2, IL-3, IL-13, IL-17, GM-CSF, and TNF was not enhanced in both infections at early time points compared to naive mice (data not shown). In contrast, serum levels of IFNα were particularly high in LCMV infected mice compared to the serum levels in MCMV infected mice (*Figure 5A*). Consistent with this, at 24 hr LCMV also induced higher expression of pro-inflammatory cytokines, which have been described to be downstream of type I IFN signaling (i.e., Rantes, IL-6, KC, Mip-1β and MCP-1) (*Teijaro et al., 2013*). However, after 48 hr the concentrations of these cytokines were comparable (*Figure 5B*). Thus, a divergent pro-inflammatory environment is induced early upon LCMV and MCMV infections.

To determine whether the high type I IFN levels that are induced during LCMV infection substitute the CD28/B7 costimulation promoting CD8$^+$ T cell expansion, we investigated the relationship between type I IFN signaling and B7-mediated costimulation in driving LCMV-specific CD8$^+$ T cell expansion. Blocking antibodies for the type I IFN receptor (IFNAR) were administered during LCMV infection and resulted in severely diminished LCMV-specific CD8$^+$ T cell responses in WT mice (*Figure 5C*). IFNAR blocking antibodies administered in *Cd80/86*$^{-/-}$ mice also severely hampered LCMV-specific responses (*Figure 5C*). Notably, the LCMV-specific CD8$^+$ T cell responses in WT mice with abrogated IFNAR signaling were comparable to those in IFNAR blocked *Cd80/86*$^{-/-}$ mice. Furthermore, no differences in IFNα levels were detected between WT and *Cd80/86*$^{-/-}$ mice (*Figure 5D*). Thus, the necessity for IFNAR signaling in the induction of LCMV-specific CD8$^+$ T cell responses does not change in the absence or presence of CD28/B7-mediated costimulation.

To examine direct effects of type I IFN-mediated signaling on CD8$^+$ T cell expansion, *Ifnar1*$^{+/+}$ and *Ifnar1*$^{-/-}$ P14 cells were adoptively transferred in WT and costimulation deficient mice that were subsequently infected with LCMV. *Ifnar1*$^{-/-}$ P14 cells transferred to WT recipients were severely hampered in expansion compared to *Ifnar1*$^{+/+}$ P14 cells (*Figure 5E*), which is consistent with previous reports (*Kolumam et al., 2005*; *Aichele et al., 2006*; *Wiesel et al., 2012*; *Crouse et al., 2014*; *Xu et al., 2014*) and confirms that type I IFNs drive directly LCMV-specific CD8$^+$ T cell expansion. *Ifnar1*$^{+/+}$ P14 cells in *Cd80/86*$^{-/-}$ mice expanded vigorously and comparable to WT host mice. Importantly, *Ifnar1*$^{-/-}$ P14 cells failed to expand in *Cd80/86*$^{-/-}$ mice as well and showed a slightly weaker expansion potential as *Ifnar1*$^{-/-}$ P14 cells in WT mice (*Figure 5E*). These data show that type I IFNs act directly on LCMV-specific CD8$^+$ T cells, and that in the absence of this signal 3 cytokine the non-dependence of B7-mediated costimulation in driving LCMV-specific T cell expansion is to some extent altered, indicating that type I IFN signaling in expanding CD8$^+$ T cells is slightly redundant with B7-mediated costimulation signals.

Next, we examined the relationship between type I IFN signaling and the B7-mediated pathway during MCMV infection. First we tested whether MCMV-specific CD8$^+$ T cell responses, which are driven by B7-mediated signals, are influenced by the type I IFN pathway. Adoptive transfer of *Ifnar1*$^{+/+}$ and *Ifnar1*$^{-/-}$ P14 cells in WT mice that were subsequently infected with MCMV-IE2-GP33 resulted in profound expansion of the *Ifnar1*$^{+/+}$ P14 cells but also of *Ifnar1*$^{-/-}$ P14 cells, although slightly diminished compared to *Ifnar1*$^{+/+}$ P14 cells. Adoptive transfer of P14 cells in *Cd80/86*$^{-/-}$ mice resulted in hampered expansion of *Ifnar1*$^{+/+}$ and even more so of *Ifnar1*$^{-/-}$ P14 cells, indicating that CD8$^+$ T cells that develop during MCMV infection are to a small degree affected by type I IFN signaling (in a somewhat redundant manner with B7-mediated costimulation) but are most critically dependent on B7-mediated signals (*Figure 5F*). Next, we examined if the B7-dependent MCMV-specific CD8$^+$ T cell response can be boosted via supplementary triggering of the type I IFN pathway. We used recombinant IFNα2 that was functional both in vitro, as determined by a cytopathic effect inhibition assay (*Figure 5—figure supplement 1A*), and in vivo as evidenced by increased expression of CD69 on lymphocytes at 18 hr upon i.p. administration (*Figure 5—figure supplement 1B*). Addition of recombinant type I IFN on day 1 and 2 during MCMV-IE2-GP33 infection in mice that received *Ifnar1*$^{+/+}$ and *Ifnar1*$^{-/-}$ P14 cells, caused no significant increase in the expansion of the P14 cells either transferred in WT or *Cd80/86*$^{-/-}$ mice, indicating that additional type I IFN signaling has negligible impact on B7-mediated signals that drive T cell expansion in MCMV infection (*Figure 5F*). Administration of recombinant type I IFN during peptide vaccination, however, improved GP33-specific CD8$^+$ T cell expansion, which indicated that IFNα is able to enhance T cell expansion in a low inflammatory context (*Figure 5G*).

To examine if the dependence of T cell expansion on B7-mediated costimulatory signals could be changed by other soluble factors than type I IFN, serum of mice that were infected for 2 days with LCMV was transferred to MCMV-infected WT and *Cd80/86*$^{-/-}$ mice. However, no differences were

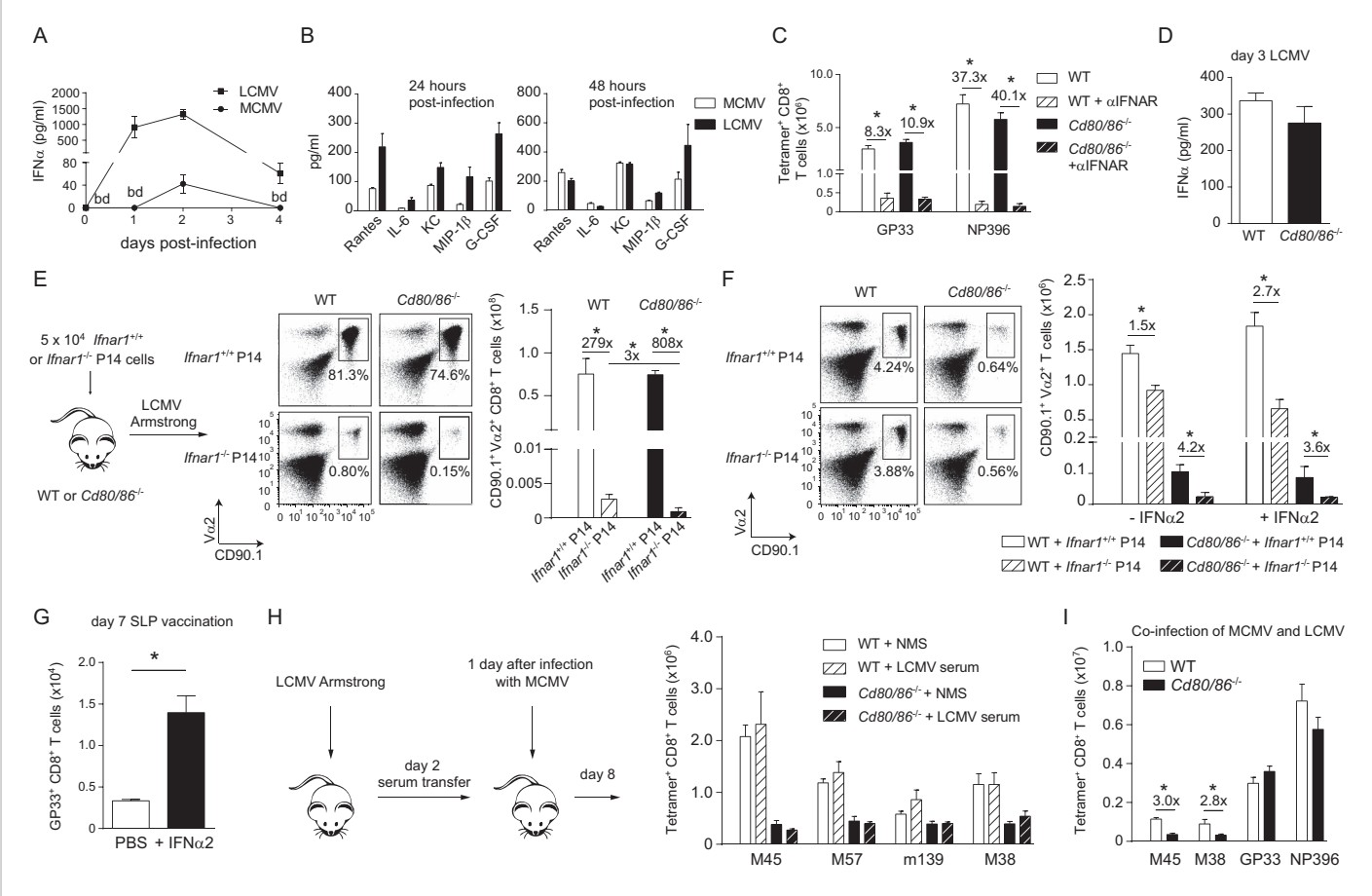

**Figure 5**. Influence of type I IFN signaling on the requirement of CD28/B7-mediated costimulation. WT mice were infected with $1 \times 10^4$ PFU MCMV-Smith or $2 \times 10^5$ PFU LCMV Armstrong and at indicated times post-infection serum was collected. (**A**) Levels of IFNα in serum in time are shown (bd = below detection limit). (**B**) Concentrations of different pro-inflammatory cytokines as determined 24 and 48 hr post-infection. (**C**) Type I interferon receptor (IFNAR) blocking antibodies were administrated during LCMV infection in WT and *Cd80/86*−/− mice. The magnitude of the virus-specific CD8+ T cell response determined by MHC class I tetramer binding at day 7 post-infection is shown. Fold difference and significance (*p < 0.05) is indicated. (**D**) IFNα levels in serum are shown 3 days post LCMV infection. (**E**) Experimental setup: $5 \times 10^4$ CD90.1+ *Ifnar1*+/+ and *Ifnar1*−/− P14 cells were adoptively transferred in WT and *Cd80/86*−/− mice that were subsequently infected with $2 \times 10^5$ PFU LCMV Armstrong. 7 days post-infection the total numbers of splenic P14 cells was determined. Representative flow cytometric plots show gated CD3+/CD8+ T cells stained for cell surface expression of CD90.1 and Vα2. Fold difference and statistical significance (*p < 0.05) between groups is indicated in the bar graphs. (**F**) Similar setup as in (**E**) except mice were infected with $1 \times 10^5$ PFU MCMV-IE2-GP33. In addition, on day 1 and 2, half of the mice received $1 \times 10^5$ units IFNα. 8 days post-infection the magnitude of the P14 cells in the spleen was determined. Representative flow cytometric plots show gated CD3+/CD8+ cells stained for cell surface expression of CD90.1 and Vα2. Bar graph shows total number of P14 cells in WT and *Cd80/86*−/− mice, and fold difference and statistical significance (*p < 0.05) between groups is indicated. (**G**) Mice were vaccinated with 75 µg SLP containing the GP33 epitope in PBS. $1 \times 10^5$ units IFNα was administrated after 18 and 48 hr. At day 7 post-vaccination, GP33-specific CD8+ T cell responses were analyzed. Significance between groups is indicated (*p < 0.05). (**H**) Experimental setup: WT mice were infected with $2 \times 10^5$ PFU LCMV Armstrong and 2 days post-infection serum was collected and transferred to mice that were infected 1 day prior with $1 \times 10^4$ PFU MCMV. The MCMV-specific CD8+ T cell response was determined 8 days post-infection by MHC class I tetramer binding. (**I**) WT and *Cd80/86*−/− mice were co-infected with $2 \times 10^5$ PFU LCMV Armstrong and $1 \times 10^4$ PFU MCMV, and virus-specific responses were analyzed 7 days post-infection by MHC class I tetramer binding. Fold difference and significance (*p < 0.05) is indicated. Data in all bar graphs are expressed as mean + SEM (n = 4–8 mice per group) of at least two independent experiments.

The following figure supplement is available for figure 5:

**Figure supplement 1**. Recombinant type I IFN is functional in vitro and in vivo.

found in the magnitude of the MCMV-specific CD8+ T cell response (*Figure 5H*), indicating that soluble factors in the LCMV environment do not enhance MCMV-specific CD8+ T cell expansion. To unequivocally demonstrate the uniqueness of the viral context to induce B7-mediated costimulation

dependence, WT mice were co-infected with MCMV and LCMV. Remarkably, during this co-infection, MCMV-specific responses were still dependent on B7-mediated signals whereas LCMV-specific CD8+ T cells were not (*Figure 5I*). Together, these data show that during an LCMV and MCMV infection a unique local environment is induced that principally determines the costimulatory requirements of the activated antigen-specific CD8+ T cells, and that direct type I IFN signaling in CD8+ T cells is slightly redundant with B7-mediated costimulation.

## Costimulatory ligands are highly expressed in LCMV infection

To further delineate factors that could locally contribute to the CD28/B7 costimulation independence of CD8+ T cell expansion during LCMV infection, we characterized the expression of cell surface bound molecules that could impact T cell expansion. First, we examined if B7 molecules were induced upon LCMV infection. Expression of both B7.1 and B7.2 was upregulated on CD11c+ and CD11b+ cells early in infection (*Figure 6A*). Strikingly, expression levels of B7.1 and B7.2 on these myeloid subsets were higher in LCMV infection as compared to MCMV infection. Thus, the non-dependence of B7-mediated costimulation for LCMV-specific CD8+ T cell expansion is not due to hampered expression of these costimulatory ligands during LCMV infection.

Besides costimulation via the CD28/B7 pathway, costimulatory signals can also be provided by TNFR superfamily members and their ligands including CD27/CD70, OX40/OX40L and 4-1BB/4-1BBL. Therefore we compared the expression of the costimulatory ligands CD70, OX40L and 4-1BBL in an LCMV and MCMV environment. Expression of both CD70 and 4-1BBL were much higher induced on CD11b+ and CD11c+ cells in LCMV infection as compared to MCMV infection (*Figure 6A,B*). Furthermore, OX40L levels were increased in LCMV infection as well, although this expression was relatively low (*Figure 6A,B*). Also compared to VV infection, elevated expression levels of all costimulatory ligands were observed on CD11b+ cells in the spleen in LCMV infection (*Figure 6—figure supplement 1A*). On CD11c+ cells, B7.2 and 4-1BBL expression was increased in LCMV infection but the levels of B7.1, CD70 and OX40L were comparable between VV and LCMV infection (*Figure 6—figure supplement 1*). The elevated costimulatory ligand expression levels found upon LCMV infection were partially associated with the high type I IFN levels within the LCMV-induced environment, as abrogation of type I IFN signaling, resulted to some extent in diminished costimulatory ligand expression (*Figure 6—figure supplement 1B*). Together these data show that in LCMV infection an overall elevated expression level of costimulatory ligands is induced, which is partially induced in a type I IFN dependent manner.

## Redundant roles for costimulatory receptor/ligand interactions in driving LCMV-specific CD8+ T cell expansion

As multiple costimulatory molecules are highly induced during LCMV infection, we hypothesized that this might lead to a redundancy of costimulatory signals to be received by the responding T cells. The TNFR superfamily member, CD27, is analogous to CD28 expressed on naive T cells, and binds the only known ligand CD70. In *Cd70*−/− mice, no significant differences were found in the magnitude of the LCMV-specific CD8+ T cell response, indicating that the CD27/CD70 costimulatory pathway by itself has a limited or redundant role during LCMV infection (*Figure 7A*). To investigate if CD70 and B7-mediated costimulation have overlapping roles in driving T cell expansion, we further examined LCMV-specific responses in mice genetically deficient for both CD70 and the B7 molecules. These *Cd70/80/86*−/− mice were viable and had no defects in the development of diverse hematopoietic cell populations (*Figure 7—figure supplement 1A–C*). Moreover, no alterations in the TCRβ repertoire were found (*Figure 7—figure supplement 1D*). Both GP33- and NP396-specific responses were significantly diminished in *Cd70/80/86*−/− mice, indicating that CD70 and B7 molecules are redundantly required for LCMV-specific CD8+ T cell expansion, and that these molecules can compensate each other (*Figure 7A*).

The redundancy of the CD27/CD70 and CD28/B7 costimulatory pathways, prompted us to further define the costimulatory requirements during LCMV infection. To determine if OX40L and 4-1BBL-mediated interactions impact LCMV-specific CD8+ T cell responses, blocking antibodies were administrated. No significant differences were found when OX40L and 4-1BBL were blocked, however, when both pathways were abrogated the magnitude of the LCMV-specific CD8+ T cell response was significantly diminished (*Figure 7A,B*). Strikingly, LCMV-specific CD8+ T cell responses

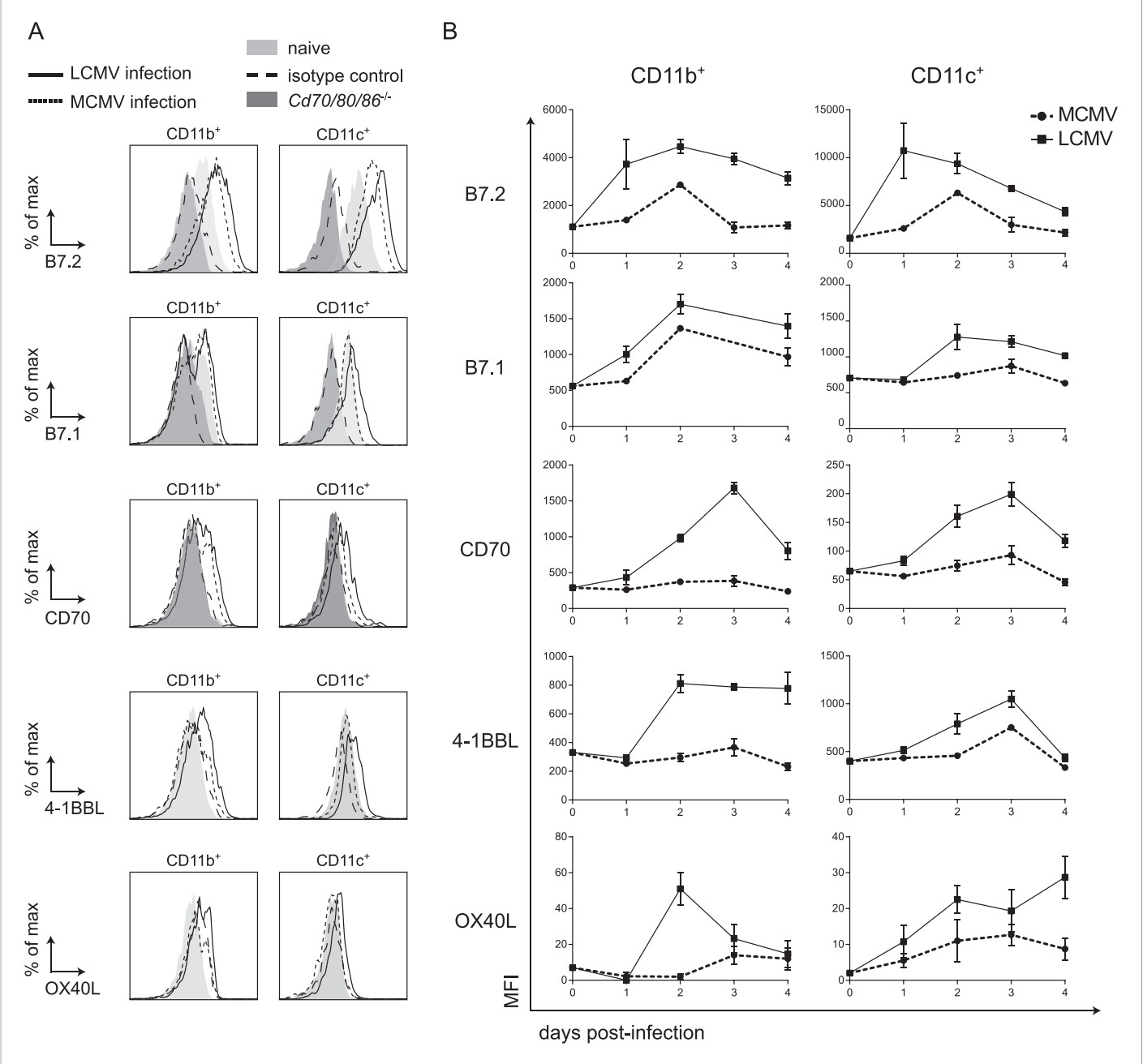

**Figure 6**. LCMV infection induces high expression of costimulatory ligands. (**A**) Mice were infected with $2 \times 10^5$ PFU LCMV Armstrong or $1 \times 10^4$ PFU MCMV and costimulatory ligand expression was determined in the spleen. Histograms show cell surface expression of indicated costimulatory molecules on CD11b+ or CD11c+ cells at day 2 post-infection with either MCMV or LCMV. Representative staining of CD11b+ and CD11c+ cells from naive WT and $Cd70/80/86^{-/-}$ mice are depicted for comparison. Staining with an isotype control antibody is indicated as well. (**B**) Graphs depict mean fluorescence intensity (MFI) of costimulatory ligand expression on CD11b+ or CD11c+ cells in time. For each sample the MFI of the corresponding isotype control was subtracted from the MFI for each costimulatory ligand. Graphs are expressed as mean ± SEM (n = 4 mice per group) of at least two independent experiments.

The following figure supplement is available for figure 6:

**Figure supplement 1**. Costimulatory ligands are highly induced in LCMV infection.

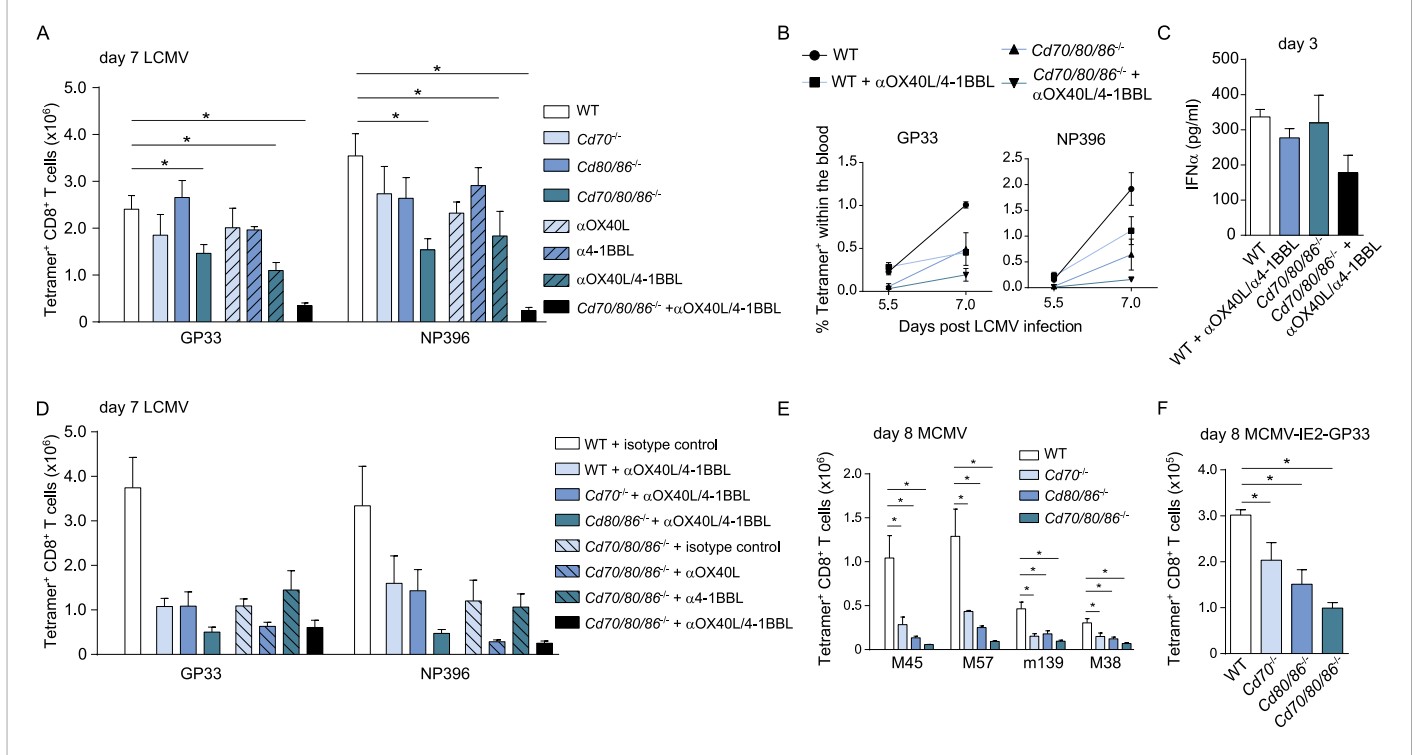

**Figure 7**. Redundant roles for costimulatory molecules in driving LCMV-specific CD8[+] T cell expansion. (**A**) WT and costimulation deficient (i.e., *Cd70*[−/−], *Cd80/86*[−/−] and *Cd70/80/86*[−/−]) mice were infected with 2 × 10^5 PFU LCMV Armstrong. OX40L and/or 4-1BBL-mediated costimulation was abrogated by administration of blocking antibodies. The LCMV-specific CD8[+] T cell response was determined 7 days post-infection using MHC class I tetramers. (**B**) The percentage of tetramer[+] CD8[+] T cells in the blood within the live gate at day 5.5 and day 7 post LCMV infection is shown as mean ± SEM. (**C**) IFNα levels in serum are shown 3 days post LCMV infection (n = 4 mice per group). (**D**) OX40L and/or 4-1BBL-mediated costimulation was abrogated by administration of blocking antibodies in WT and costimulation deficient mice that were subsequently infected with 2 × 10^5 PFU LCMV Armstrong. The LCMV-specific CD8[+] T cell response was determined 7 days post-infection using MHC class I tetramers. All responses in mice receiving blocking antibodies to costimulatory molecules were significantly decreased (p < 0.05) compared to WT mice receiving isotype control antibodies. (**E**) The magnitude of splenic MCMV-specific CD8[+] T cell pools determined by MHC class I tetramer staining after infection with 1 × 10^4 PFU MCMV-Smith is indicated. (**F**) The magnitude of the splenic GP33-specific CD8[+] T cell response at day 8 post-infection with 1 × 10^5 PFU MCMV-IE2-GP33 is shown. All data in bar graphs are expressed as mean + SEM (n = 5–12 mice per group of at least two independent experiments; *p < 0.05).

The following figure supplements are available for figure 7:

**Figure supplement 1**. *Cd70/80/86*[−/−] mice have no defects in development of different hematopoietic populations.

**Figure supplement 2**. OX40L- and 4-1BBL-mediated costimulation is dispensable for primary expansion of MCMV-specific CD8[+] T cells.

were drastically decreased when OX40L/4-1BBL blockade was performed in mice lacking CD70 and B7-mediated costimulation. This diminished response was not due to defective induction of type I IFN, as IFNα levels in the serum of these mice were not substantially altered compared to WT mice (*Figure 7C*). We further delineated the redundancy between different costimulatory molecules by additionally blocking OX40L- and/or 4-1BBL-mediated interactions in *Cd70* and *Cd80/86* deficient mice. Dual blockade of OX40L and 4-1BBL in *Cd80/86*[−/−] mice, and OX40L blockade in *Cd70/80/86*[−/−] mice showed comparable responses to mice in which all costimulatory pathways were abrogated, indicating that the most pronounced effects on LCMV-specific CD8[+] T cell expansion are found when both B7 and OX40L-mediated interactions are abrogated (*Figure 7D*). Together, these data indicate that virus-specific CD8[+] T cell responses during LCMV infection critically depend on a plethora of costimulatory signals that are individually dispensable because they function in a highly redundant manner.

To determine if costimulatory molecules were similarly working in a redundant manner in MCMV infection, WT and costimulation deficient mice were infected with MCMV. MCMV-specific

CD8$^+$ T cell responses in *Cd70$^{-/-}$* and *Cd80/86$^{-/-}$* mice were significantly diminished, however responses in *Cd70/80/86$^{-/-}$* mice were even lower (*Figure 7E*), indicating both a non-redundant and cooperative role for CD70 and B7-mediated costimulation in driving MCMV-specific T cell expansion. Similar results were obtained for GP33-specific CD8$^+$ T cell responses using MCMV-IE2-GP33 (*Figure 7F*). Abrogation of OX40L or 4-1BBL-mediated signals upon MCMV infection has been shown to minimally impact the initial expansion of MCMV-specific CD8$^+$ T cells (*Humphreys et al., 2007*, *2010*). Moreover, we found that upon dual blockade of OX40L and 4-1BBL-mediated interactions MCMV-specific T cell responses were not affected as well (*Figure 7—figure supplement 2*). These results indicate that redundancy between different costimulatory molecules is induced by the viral context.

## Type I IFN signaling in viral-specific CD8$^+$ T cells is slightly redundant with costimulatory signals

In both MCMV and LCMV infection, the virus-specific CD8$^+$ T cell response is more affected in the absence of both CD70 and B7-mediated costimulation as compared to mice lacking only one of these costimulatory pathways. We next determined if type I IFN signaling is altered upon abrogation of dual CD70 and B7-mediated costimulation. Similar to what is found for endogenous LCMV-specific CD8$^+$ T cell responses, *Ifnar1$^{+/+}$* P14 cells expanded well in WT, *Cd70$^{-/-}$* and *Cd80/86$^{-/-}$* mice and were to some extend hampered in expansion in *Cd70/80/86$^{-/-}$* mice (*Figure 8A*). The *Ifnar1$^{-/-}$* P14 cells were rigorously hindered in their expansion, when transferred in WT mice and even more so in costimulation deficient mice. This reduced expansion of the *Ifnar1$^{-/-}$* P14 cells in the costimulation deficient mice as compared to WT mice indicates slight redundancy of type I IFN signaling with costimulatory-driven signals in expanding CD8$^+$ T cells.

Furthermore, *Ifnar1$^{+/+}$* P14 cells were transferred to mice that were infected with MCMV-IE2-GP33. In this setting, P14 cell expansion was critically dependent on both CD70- and B7-mediated costimulation (*Figure 8B*). Compared to *Ifnar1* proficient P14 cells, *Ifnar1* deficient P14 cells had a higher degree of type I IFN dependence in the absence of costimulation, which was most pronounced when both CD70 and B7 costimulatory molecules were lacking (*Figure 8B*). Thus, type I

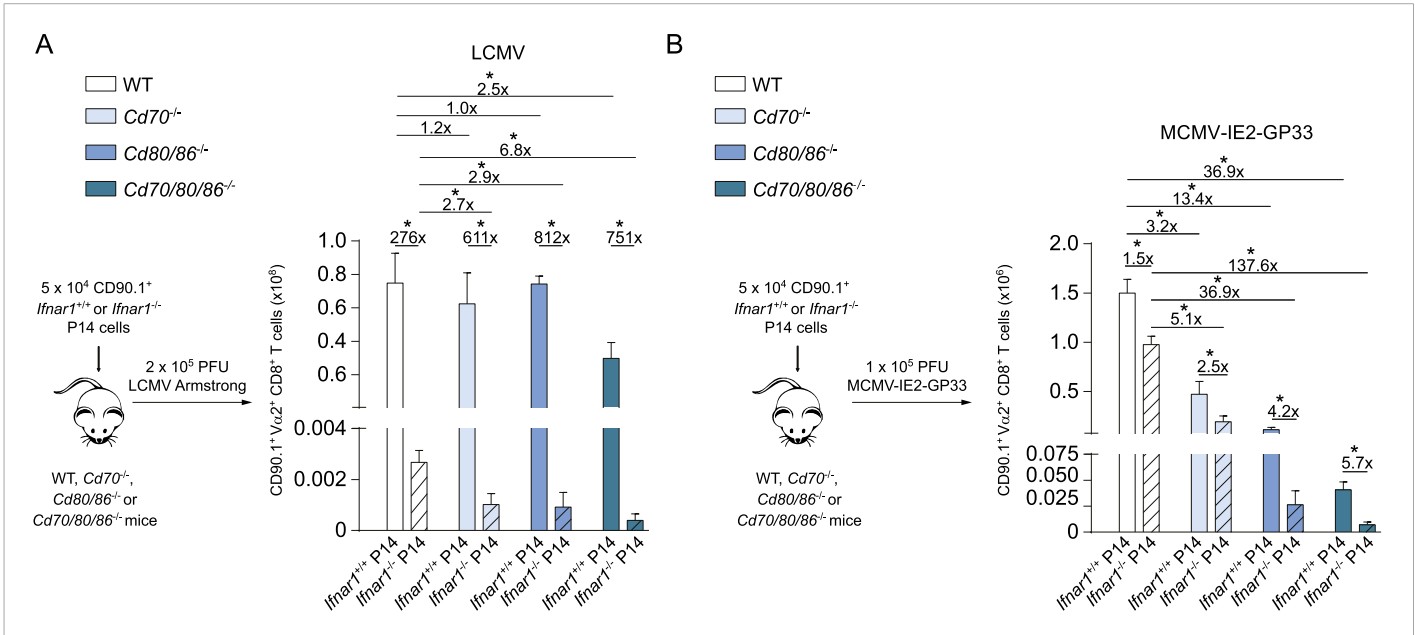

**Figure 8.** Type I IFN signaling in viral-specific CD8$^+$ T cells is slightly redundant with costimulatory signals. (**A**) Schematic of experimental setup: *Ifnar1$^{+/+}$* and *Ifnar1$^{-/-}$* P14 cells were adoptively transferred in WT, *Cd70$^{-/-}$*, *Cd80/86$^{-/-}$* and *Cd70/80/86$^{-/-}$* mice that were subsequently infected with 2 × 10$^5$ PFU LCMV. 7 days post-infection the total numbers of P14 cells was determined in the spleen. (**B**) Similar setup as in (**A**) except mice were infected with 1 × 10$^5$ PFU MCMV-IE2-GP33. 8 days post-infection the magnitude of the P14 cells was determined. Data in bar graphs are expressed as mean + SEM (n = 4–8 mice per group) and representative of two independent experiments. The fold difference and significance (*p < 0.05) is indicated.

IFNs have a slight stimulating activity for CD8+ T cells in MCMV infection, which is more pronounced in the absence of CD70 and B7-mediated signaling, indicating that also during MCMV infection partial redundancy of type I IFN signaling with costimulation during CD8+ T cell expansion occurs.

## Discussion

Determining the critical components required for T cell expansion in a given situation is of utmost importance for understanding resistance to virus infections and improving vaccination strategies. Using different viral models we show that the pathogen-induced environment dictates the utilization of costimulatory signals that drive CD8+ T cell expansion.

Primary LCMV-specific CD8+ T cell responses have long been considered to be costimulation independent (*Shahinian et al., 1993*; *Kundig et al., 1996*; *Andreasen et al., 2000*; *Grujic et al., 2010*; *Eberlein et al., 2012*). Nevertheless, the development of LCMV-specific memory CD8+ T cell formation is hampered during *Cd28* or *Cd80/86* deficiency (*Grujic et al., 2010*; *Eberlein et al., 2012*), indicating that CD28/B7-mediated costimulation occurs during LCMV infection, which is in agreement with our study. We also found that the CD27/CD70 pathway has negligible costimulatory effects for LCMV-specific CD8+ T cell expansion when solely this pathway is abrogated. This has been observed by others as well (*Matter et al., 2005*; *Schildknecht et al., 2007*), but recent reports suggested that blockade of the CD27/CD70 pathway can to some extend impair CD8+ T cell responses during acute LCMV infection (*Penaloza-Macmaster et al., 2011*; *Munitic et al., 2013*). Importantly, here we show that LCMV-specific CD8+ T cell responses are in fact critically dependent on costimulatory signals, but these signals operate in a highly redundant manner in which both members of the costimulatory CD28/B7 family and TNFR/TNF family take part.

The overall expression of costimulatory ligands in the LCMV milieu exceeded the expression levels found upon an MCMV or VV infection. In this respect, it is of interest to note that abrogation of exclusively the CD28/B7 or the CD27/CD70 pathway severely hampers MCMV- and VV-specific CD8+ T cell responses (*Arens et al., 2011b*; *Salek-Ardakani et al., 2011*; *Welten et al., 2013b*), indicating that in these infections the costimulatory molecule levels are likely limited leading to non-redundant roles of costimulatory molecules. Unhampered LCMV-specific responses are observed upon dual 4-1BBL and CD28 abrogation (*DeBenedette et al., 1999*) and this is consistent with our data showing that multiple pathways than these have to be abrogated to observe diminished LCMV-specific CD8+ T cell responses virus-specific responses. The higher expression levels of costimulatory ligands within the LCMV environment is likely causing the redundancy amongst CD28/B7 and TNFR/TNF family members in driving LCMV-specific T cell expansion. Of interest is that even further improvement of B7-mediated signaling due to CTLA-4 blockade did not advance LCMV-specific CD8+ T cell expansion, suggesting that the observed higher expression of costimulatory molecules is at a maximal level with respect to stimulating T cells.

Strong replicating VV-strains employ more costimulatory receptors as compared to weak replicating VV-strains (*Salek-Ardakani et al., 2011*). Furthermore, 4-1BBL-mediated interactions are critical during severe influenza virus infections but dispensable upon a mild influenza virus (*Lin et al., 2009*), indicating that the strength of the inflammatory environment dictates the employment of different costimulatory receptors. Given the higher costimulatory molecule expression, one could argue that LCMV infection elicits an elevated inflammatory milieu as compared to most other infections. Consistent with this notion is that in LCMV infection very high levels of type I IFNs are induced, which are partly responsible for the high costimulatory ligand expression. An elevated expression of costimulatory molecules in LCMV infection might also be related to a lack of immunomodulatory effects that dampen costimulatory molecule expression. During MCMV infection for example, the B7.1 and B7.2 expression in virus-infected cells is downmodulated by the virus by sophisticated immune evasion mechanism (*Loewendorf et al., 2004*; *Mintern et al., 2006*; *Arens et al., 2011a*). Perhaps related to this, is that the CD8+ T cell response to MCMV is predominantly mediated by cross-priming APCs, which are by definition not directly infected by the virus (*Torti et al., 2011*; *Busche et al., 2013*).

Shared signaling pathways might underlie the observed redundancy among members of the costimulatory TNFR family and CD28 family. TNFR family members are known to signal via TRAF molecules, which are coupled to the activation of the NF-κB pathway via both the canonical and the non-canonical routes (*Croft, 2009*). CD28 is also able to signal via the NF-κB route (*Boomer and Green, 2010*). Another shared signaling pathway of CD28 and TNFR family members might be the c-Jun kinase pathway, which is coupled to proliferation as well (*Gravestein et al., 1998*; *Skanland et al., 2014*).

We found redundancy between CD28 and CD27 signaling on CD8[+] T cell expansion in MCMV and LCMV infection, and this has been found in influenza virus infection as well (*Hendriks et al., 2003*). However, besides redundancy between CD28/B7 and TNFR/TNF families also redundancy among costimulatory TNFR family members likely happened as the response was most compromised in settings where multiple TNFR family members were targeted. The latter is consistent with observations in the influenza virus infection model, where virus-specific T cells that accumulate in the lung but not in the spleen were collectively dependent on signals mediated via a variety of TNFR family members (*Hendriks et al., 2005*).

We found a prominent role for the pathogenic milieu in directing CD8[+] T cell responses and dictating the requirements for certain costimulatory signals. The fact that even upon LCMV and MCMV co-infection the costimulatory requirements for T cell expansion are not altered, suggest that this instruction occurs locally, likely at the level of APC-T cell interaction. The majority of the MCMV-specific CD8[+] T cells is activated via cross-priming (*Torti et al., 2011*; *Busche et al., 2013*), and whether both direct and cross-priming occur during LCMV infection is unclear (*Freigang et al., 2007*). Nevertheless CD11c[+] APCs are critical for LCMV-specific CD8[+] T cell priming (*Probst and van den Broek, 2005*). Moreover, because of different tropisms it is unlikely that MCMV and LCMV co-infect the very same cells and that the viral epitopes are presented by the same APC (*Matloubian et al., 1993*; *Alexandre et al., 2014*). Since APCs need to be directly activated for adequate T cell priming rather than by environmental inflammatory signals (*Kratky, 2011*), our data are consistent with a scenario where the two viruses activate APCs in a different manner resulting in differential provision of costimulatory signals. The enhanced costimulation during LCMV infection may besides due to stronger and distinctive (local) inflammation also be a consequence of longer and/or stronger antigen-presentation as compared to other viral infections. However, LCMV and MCMV are both natural mouse pathogens and infection with these viruses results in virus levels that peak around day 4 post-infection in the spleen and liver (*Buchmeier et al., 1980*; *Cicin-Sain et al., 2008*). Nevertheless, differential kinetics of antigen-presentation of the viral epitopes is possible.

Perhaps related to our results are the observations that the pathogen-specific inflammatory environment dictates the fate of responding CD8[+] T cells allowing shaping of effector and memory T cell formation (*Obar et al., 2011*; *Keppler et al., 2012*; *Plumlee et al., 2013*). This may be connected with pathogen-specific tuning of the antigen-sensitivity of CD8[+] T cells by enhancing TCR signaling (*Richer et al., 2013*), the induction of distinct inflammatory cytokine levels (*Thompson et al., 2006*) and/or by instructing the costimulatory pathway usage (our results). Although in vitro the requirements for CD28/B7-mediated costimulation can differ for primary and memory cells (*Flynn and Mullbacher, 1996*), we found in vivo that CD28/B7-mediated costimulation was important for the expansion of both naive and memory CD8[+] T cells in MCMV infection. This is consistent with models of influenza virus, VV and murine γ-herpesvirus (*Borowski et al., 2007*; *Fuse et al., 2008*) that require B7-mediated signals for primary and secondary expansion of virus-specific CD8[+] T cells. However, the APCs that prime memory vs naive T cells might differ (*Belz et al., 2007*).

Type I IFNs are not required for the expansion of human memory CD8[+] T cells in vitro (*Hervas-Stubbs et al., 2010*). In experimental in vivo models, however, the inflammatory environment determines the signal 3 (i.e., type I IFN and IL-12 signaling) dependency upon secondary infection independent of the context of priming (*Keppler and Aichele, 2011*). Correspondingly, we observed that the milieu of the infectious pathogen during the recall response determines the requirements for costimulatory signals as well, and suggests that the responsiveness of T cells during the initial expansion is plastic and can be modified during antigenic re-challenge.

Collectively, our results highlight the importance of the inflammatory environment for both primary and secondary CD8[+] T cell expansion. These findings can be beneficial for pre-clinical exploration of adoptive T cell settings, where antigen-specific T cells are expanded to large numbers. In addition, our report has important implications for prime-boost vaccination strategies, as it provides evidence for the plasticity of memory T cells that is shaped by the nature of the pathogen to generate them.

## Materials and methods

### Mice

C57BL/6 mice were obtained from Charles River and were used as WT mice. *Cd70*[−/−] (*Coquet et al., 2013*), *Cd80/86*[−/−] (*Borriello et al., 1997*) and *Ptprc*[a] (*Cd45.1, Ly5.1*) mice were bred in house to the

obtained C57BL/6 background. *Cd70/80/86*−/− mice were generated by crossing *Cd70*−/− with *Cd80/ 86*−/− mice. All animals were maintained on specific pathogen free conditions at the animal facility in Leiden University Medical Center (LUMC). Mice were matched for gender and were between 8-12 weeks at the start of each experiment. IFNAR proficient (*Ifnar1*+/+) and deficient (*Ifnar1*−/−) P14 TCR transgenic mice on a CD90.1+ C57BL/6 background were generated by breeding as described (*Keppler et al., 2012*). All animal experiments were approved by the Animal Experiments Committee of LUMC (reference numbers: 12,006, 13,150, 14,046 and 14,066) and performed according to the recommendations and guidelines set by LUMC and by the Dutch Experiments on Animals Act that serves the implementation of 'Guidelines on the protection of experimental animals' by the Council of Europe.

## Pathogens and infections

MCMV-Smith was obtained from the American Type Culture Collection (Manassas, VA). Stocks were derived from salivary glands of infected BALB/c mice as described elsewhere (*Schneider et al., 2008*). Viral titers were determined as described (*Welten et al., 2013b*). For an in vivo MCMV infection, mice were infected intraperitoneal (i.p.) with $1 \times 10^4$ PFU MCMV-Smith. To generate MCMV-IE2-GP33, MCMV-M45-GP33 and MCMV-M45-SIINFEKL, nucleotide sequences encoding the $GP_{33-41}$ epitope (GP33) of LCMV or the SIINFEKL epitope of chicken ovalbumin were inserted by targeted mutagenesis at the C-terminus of the M45 or IE2 genes, directly in front of the stop codon. Two alanine residues in front of the peptide sequences were placed in order to enhance proteasomal cleavage. Recombinant virus was reconstituted as described elsewhere (*Dekhtiarenko et al., 2013*). Mice were infected i.p. with $1 \times 10^5$ PFU MCMV-IE2-GP33, MCMV-M45-GP33 or MCMV-M45-SIINFKEL.

LCMV Armstrong was propagated on BHK cells. The titers were determined by plaque assays on Vero cells as described (*Ahmed et al., 1984*). For LCMV Armstrong infection, mice were infected i.p. with $2 \times 10^5$ PFU (high dose) or $2 \times 10^2$ PFU (low dose). For co-infection experiments, mice were infected with $2 \times 10^5$ PFU LCMV and $1 \times 10^4$ PFU MCMV-Smith. VV strain WR was purchased from the American Type Culture Collection, grown on HELA cells and quantified on VeroE6 cells as described (*Davies et al., 2005*). Mice were infected i.p with $2 \times 10^5$ PFU VV. *L. monocytogenes* (LM) expressing GP33 and the attenuated LM-Quadvac strain expressing four epitopes of VV (i.e., A24R, C4L, K3L and B8R) and the SIINFEKL epitope of OVA are described elsewhere (*Zenewicz et al., 2002*; *Lauer et al., 2008*). Mice were challenged intravenously (i.v.) with $1.5 \times 10^3$ CFU LM-GP33 or with $1 \times 10^6$ CFU LM-Quadvac.

## In vivo antibody treatment

For blockade of IFNAR, mice received 1 mg of IFNAR blocking antibody (clone MAR1-5A3; Bio X Cell, West Lebanon, NH, United States) on day −1 and 1 post-infection. For blockade of CTLA-4, 200 μg of αCTLA-4 (clone UC10-4F10-11) was administrated on day −1, 1 and 3. For blockade of OX40L and 4-1BBL, 150 μg of αOX40L (clone RM134L) or α4-1BBL (clone TKS-1) (both Bio X Cell) or a combination of both antibodies was administrated on day −1, 1 and 3 post-infection. Control mice received a similar amount of a rat IgG isotype control antibody (clone GL113). All antibodies were administrated i.p. in 400 μl PBS.

## Flow cytometry

Splenocytes were obtained by mincing the tissue through a 70 μm nylon cell strainer (BD Biosciences, San Jose, CA, United States). Blood was collected via the tail vein. Erythrocytes were lysed in a hypotonic ammonium chloride buffer. Determination of the antigen-specific T cell response by MHC class I tetramers and intracellular cytokine staining was performed as described (*Arens et al., 2011b*). Briefly, single-cell suspensions were incubated with fluorescently conjugated antibodies and tetramers for 30 min at 4°C. To determine the expression of costimulatory ligands, spleens were first injected with 1 mg/ml collagenase and 0.02 mg/ml DNAse in IMDM without FCS, after which the spleens were chopped in small pieces and incubated for 25 min at RT. Subsequently 0.1 M EDTA was added and cells were transferred through a 70 μm cell strainer to make single cell suspensions. Next, cells were pre-incubated with normal mouse serum and Fc-block (clone 2.4G2), after which biotinylated or fluorochrome conjugated antibodies were added. For analysis of intracellular cytokines, cells were

restimulated for 5 hr with 1 µg/ml MHC class I restricted peptides in the presence of 1 µg/ml brefeldin A, followed by cell surface staining and intracellular staining for IFN-γ. The following fluorescently conjugated antibodies were purchased by BD Biosciences, eBioscience (San Diego, CA, United States) or BioLegend (San Diego, CA, United States): CD3 (V500), CD8 (A700), CD11b (eFluor450), CD11c (eFluor780), CD44 (eFluor450) CD45.1 (FITC), CD62L (eFluor780), CD70 (biotin), CD80 (FITC), CD86 (PE), CD90.1 (FITC), IFN-γ (APC), KLRG1 (PE-Cy7), OX40L (biotin), Vα2 (PE), 4-1BBL (biotin). Fluorochrome-conjugated streptavidin (PE, APC or Brilliant Violent 605) was used to detect biotinylated antibodies. Flow cytometric acquisition was performed on a BD LSR II and cells were sorted using a BD FACSAria. Data were analyzed using FlowJo software (TreeStar, Ashland, OR, United States).

## MHC class I tetramers and peptides

MHC class I $D^b$ restricted tetramers for the $OVA_{257-264}$ epitope (SIINFEKL), the MCMV epitope $M45_{985-993}$, the LCMV epitopes $GP_{33-41}$ and $NP_{396-404}$, and MHC class I $K^b$ restricted tetramers for the MCMV epitopes $M57_{816-824}$, $m139_{419-426}$, and $M38_{316-323}$, and the VV epitopes $B8R_{20-27}$ and $A3L_{270-277}$ were produced as described (Altman et al., 1996). The following class I-restricted peptides were used: $M45_{985-993}$, $m139_{419-426}$, $M38_{316-323}$, $GP_{33-41}$ and $NP_{396-404}$. The following SLP containing the GP33 epitope (underlined) was used for vaccination: VITGIKAVYNFATCGIFALIS. Mice were vaccinated at the tail base with 75 µg SLP in PBS either combined with 20 µg CpG or supplemented with $1 \times 10^5$ units IFNα injected i.p. in 200 µl PBS at 18 and 48 hr post-vaccination.

## Multiplex assay

Blood was collected retro-orbitally and clotted for 30 min. Serum was collected after centrifugation and stored at −80°C until further use. Cytokines were measured in serum using a mouse Bio-Plex Pro Mouse Cytokine 23-plex immunoassay (Bio-Rad, Herculus, CA, United States) according to manufacturer's protocol. IFNα was measured with a mouse ProcartaPlex multiplex immunoassay (eBioscience).

## Adoptive transfer experiments

Splenic $Ifnar1^{+/+}$ and $Ifnar1^{-/-}$ CD90.1 P14 cells were enriched by negative selection of CD8$^+$ T cells (BD Biosciences) and $5 \times 10^4$ cells were adoptively transferred in WT and costimulation deficient mice that were subsequently infected with either $2 \times 10^5$ PFU LCMV Armstrong or $1 \times 10^5$ PFU MCMV-IE2-GP33. 7 days post LCMV or 8 days post MCMV infection the magnitude of P14 cells was determined.

For adoptive transfer of memory GP33-specific CD8$^+$ T cells, CD45.1$^+$ congenic mice were infected with $2 \times 10^5$ PFU LCMV Armstrong. After 4 months GP33-specific memory CD8$^+$ T cells were FACS sorted using MHC class I tetramers and $2 \times 10^3$ cells were adoptively transferred into WT and $Cd80/86^{-/-}$ mice that were subsequently infected with $2 \times 10^5$ PFU LCMV Armstrong or $1 \times 10^5$ PFU MCMV-IE2-GP33. 6 days post adoptive transfer, the total number of CD45.1$^+$ GP33-specific CD8$^+$ T cells was determined. Similar experiments were performed with CD45.1$^+$ congenic mice infected with $1 \times 10^5$ PFU MCMV-IE2-GP33.

For serum transfer, WT mice were infected with $2 \times 10^5$ PFU LCMV Armstrong and after 2 days, serum was collected and 150 µl was transferred i.p. to mice that were infected 1 day before with $1 \times 10^4$ PFU MCMV-Smith. 8 days post MCMV inoculation, MCMV-specific CD8$^+$ T cell responses were determined in the spleen.

## Recombinant type I IFN

DNA encoding mouse IFNα2, the $Ifna2$ gene, was synthetically made and codon optimized by Geneart (Thermo Fisher Scientific, Waltham, MA, United States). The gene was subcloned by Gateway technology (Thermo Fisher Scientific) in pDEST17, which has an N-terminal histidine tag. After overproduction the protein was purified as described (Franken et al., 2000) and lyophilized. 2.5 mg of protein was resuspended in 1 ml 100 mM Tris HCl, 8 M Urea pH 8.0. The dissolved protein was refolded in 50 ml 0.4 M L-arginine, 100 mM Tris HCl, 2 mM EDTA, 0,5 mM oxidized glutathione, 5 mM reduced glutathione, 5% glycerol and 0.5 tablet of Complete pH 8.0. After 5 days of incubation at 10°C the solution was concentrated on an Ultracel 10 kD filter (Merck Millipore (Billerica, MA, United States)). The concentrated protein was loaded on a PBS equilibrated Hi-Load 16/60 superdex 75 column. The collected peak of the protein was concentrated on the Ultracel 10 kD filter and stored with 16% glycerol at −80°C. Protein concentration was determined by Bradford and OD$_{280}$ nm.

The bioactivity was determined according to a protocol described elsewhere (*Seeds and Miller, 2011*) with slight alterations. In short; L929 cells were seeded in 96-well plates in serum free RPMI and incubated at 37°C, the next day different dilutions of IFNα2 were added. The following day Mengovirus was added and after 2 days of incubation an MTT assay was performed (Trevigen, Gaithersburg, MD, United States). Cell survival was determined by the following formula: ($OD_{570-655}$ sample with IFNα2 and virus/$OD_{570-655}$ without virus and IFNα2) × 100%. One unit of IFNα2 was defined as the concentration at which 50% of the cytopathic effect was inhibited. Our batch had a bioactivity of $1 \times 10^6$ units/ml. For in vivo administration, mice received $1 \times 10^5$ units i.p. upon CMV infection or post vaccination.

## Statistical analysis

GraphPad Prism 6.0 software (GraphPad Software, La Jolla, CA, United States) was used for statistical analyses. To determine statistical significance between two groups an unpaired Student's *t*-test was performed. To evaluate significance between more than two groups, one-way ANOVA was used and values were compared to WT mice. Dunnett's post-hoc test was performed to correct for multiple comparisons. p-values <0.05 were considered as significant.

## Acknowledgements

We would like to thank Dr M Kikkert for kindly providing us L929 cells and Mengovirus, Edwin de Haas for cell sorting, and Els van Beelen for assistance with luminex assays.

## Additional information

### Funding

| Funder | Grant reference | Author |
| --- | --- | --- |
| Leids Universitair Medisch Centrum | Gisela Thier | Ramon Arens |

The funder had no role in study design, data collection and interpretation, or the decision to submit the work for publication.

### Author contributions

SPMW, RA, Conception and design, Acquisition of data, Analysis and interpretation of data, Drafting or revising the article; AR, Acquisition of data, Analysis and interpretation of data; KLMCF, Analysis and interpretation of data, Contributed unpublished essential data or reagents; JDO, Acquisition of data, Contributed unpublished essential data or reagents; FO, CJMM, Conception and design, Drafting or revising the article; LČ-Š, PA, Drafting or revising the article, Contributed unpublished essential data or reagents

### Ethics

Animal experimentation: Animal experiments were approved by the Animal Experiments Committee of the LUMC (reference numbers: 12006, 13150, 14046 and 14066) and performed according to the recommendations and guidelines set by the LUMC and by the Dutch Experiments on Animals Act that serves the implementation of 'Guidelines on the protection of experimental animals' by the Council of Europe.

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
