## [Decision Letter]

Thank you for sending your work entitled “The viral context instructs the requirement for costimulation in driving CD8^+^ T cell expansion” for consideration at *eLife*. Your article has been evaluated by Tadatsugu Taniguchi (Senior Editor) and two reviewers, one of who is a member of our Board of Reviewing Editors.

The Reviewing Editor and the other reviewer discussed their comments before reaching this decision, and the Reviewing Editor has assembled the following comments to help you prepare a revised submission.

The manuscript as it stands provides data of relevance in analysing the complexity of effective immune recognition pathways in quasi-real-life infection situations and is thus of clear value to journals devoted to infection and immunology. However, there are a number of issues that would need to be dealt with, using substantive newly generated data, for the manuscript to make a rigorous point broad enough to be of interest to the cross-disciplinary readership of *eLife*.

1) The part of this study that simply demonstrates contextual variations of costimulatory requirements in viral infections is not novel but conceptually similar to several previous studies, and it would be appropriate to acknowledge them. Thus, for example, the finding that LCMV responses are independent of CD28 (and by inference B7.1, B7.2) for which the original reference (Shahinian et al., Science, 1993) also noted that costimulation requirements are viral context-dependent, or the finding that anti-LCMV but not anti-influenza CD8 T cell responses are unimpaired in the absence of both CD28 and 4-1BBL (DeBenedette et al., JI, 1999), or the epitope repertoire-independence of viral infection contextual costimulation requirements (Lin et al., JI, 2009, and Salek-Ardakani 2011). Thus, the manuscript should be a bit more balanced in acknowledging such previous contributions to the field, particularly when the idea of contextual variation in costimulation requirements has been reviewed recently (Wotzman et al., Immunol Rev 2013).

2) The primary limitation of the manuscript is the absence of a comprehensive and systematic framework at least delineating if not explaining the shifting matrix of costimulatory requirements/redundancies in distinct infection/immunisation situations in vivo. Thus, at least for the pair of LCMV (Armstrong) and MCMV, an experiment of the kind shown in Figure 7, but with a more complete set of combinations, and comparing LCMV and MCMV, would be very useful. Such experiments will be best done using transferred p14-TCR-transgenic CD8 T cells to ‘control’ for TCR-repertoire-related differences and to allow the use of IFNIR^+/+^ and IFNIR^-/-^ p14 CD8 T cells to enable the inclusion of IFN-I-signaling in the analytical matrix.

Technically, too, this most novel aspect of the study is somewhat underdeveloped. For example, the blocking studies in Figure 7, show only one time point for the readout and would be more convincing if more than one time point were examined. The authors administer very large amounts of antibodies without indicating any isotype controls, raising worries about the specificity of the blockade and about possible FcR mediated APC effects. Figure 7 is a critical experiment, but it’s hard to determine how strong the result is – it would be useful to see each mouse shown as a single point and all the mice included, rather than just showing a bar graph.

3) It is necessary to explore mechanistic explanations for the inability of IFN-I (or LCMV-infected mouse day 2 serum) to allow MCMV-GP33-specific P14 CD8 T cells to become *Cd80/86* independent (Figure 5). A GP33-negative LCMV-co-infection (instead of recombinant IFN-I or LCMV-infected mouse serum) might, for example, be useful to test. If there is no reconstitution of *Cd80/86* independence for MCMV-immunisation even so, then alternative complex explanatory interactions, possibly requiring the inclusion of antigen presentation kinetics and/or infected/presenting APC-specific issues, will need to be considered and addressed substantively.

In this context, while the authors show that LCMV infected mice have much higher levels of type I IFN as well as much higher levels of costimulatory molecules than MCMV infected mice, they do not explore the relationships involved any further. Does type I IFN regulate the levels of the costimulatory molecules? For example, does IFNAR blockade lower any of the costimulatory molecules in Figure 6, or is the differential induction in MCMV versus LCMV infection due to direct response to the viral particles? Some efforts to gain insights into the ‘contextual’ role would add value.

4) As noted above, data with transferred P14-TCR-transgenic CD8 T cells in all contexts tested would be needed to address the question of whether costimulatory requirement distinctions in LCMV, MCMV, LM and SLP contexts are independent of differences, if any, in the GP33 peptide-specific TCR repertoires.

5) It would be preferable to test all conclusions carefully in situations where the extent of CD8 T cell expansion is similar between the LCMV- versus MCMV-groups being compared, since differences in the degree of expansion may have contributory consequences not apparent in the situation Figure 1.

6) It is somewhat surprising, given the importance of littermate controls, that the manuscript uses ‘wild-type’ from a commercial source as a control for in-house knockout mice. It would be useful to have some clarification of this issue.

7) The data (Figure 4) suggest that there are no differences between co-stimulatory requirements for primary versus secondary CD8 T cells. There is some controversy in the extant literature regarding the co-stimulation requirements of ‘memory’ CD8 T cells (25; 12). In fact, ‘memory’ CD8 T cells may not respond as well as naive CD8 T cells (9), and type I interferons may negatively affect memory CD8 T cell proliferation (33) under certain circumstances. If the present manuscript is to address co-stimulatory requirements of memory CD8 T cells, it would be appropriate to have these issues acknowledged and addressed.

8) The title does not reflect the novel aspects of the study; it would be useful for the title to reflect that the new finding is the redundancy of costimulatory pathways for CD8 T cell responses to LCMV, rather than simply the context-dependent nature of costimulation.

[Editors' note: further revisions were requested prior to acceptance, as described below.]

Thank you for resubmitting your work entitled “The viral context instructs the redundancy of costimulatory pathways in driving CD8^+^ T cell expansion” for further consideration at *eLife*. Your revised article has been favorably evaluated by Tadatsugu Taniguchi (Senior Editor), a Reviewing Editor, and one reviewer. The manuscript has been improved but there are some remaining issues that need to be addressed before acceptance, as outlined below: This paper elegantly shows that it is the viral context rather than the TCR repertoire or specific epitope that determines B7/CD28 dependence of CD8 T cell responses to viruses. By using LCMV epitopes or GP33 specific T cells in the context of an LCMV or MCMV infection, the authors show that LCMV induced responses are *Cd80/86* independent whereas MCMV responses are *Cd80/86* dependent. This is true of secondary responses to LCMV or MCMV delivered epitopes also, regardless of whether cells were primed by LCMV or MCMV. As LCMV infection induces particularly high levels of type I Interferon, the authors investigated the role of type I interferon in the *Cd80/86* dependence. However, blocking type I interferon receptor similarly impaired WT or *Cd80/86* knockout mice, arguing that type I interferon does not explain *Cd80/86* independence of the LCMV specific response. Neither transfer of serum from LCMV infected mice nor coinfection changed the results. LCMV infection was found to induce higher levels of several costimulatory molecules and blocking and knockout mice were used to show redundancy of costimulatory pathways for LCMV specific CD8 T cell responses.

A remaining question is: does LCMV induce more costimulation through more inflammation or more antigen? It would be of interest to determine the level of viral antigen in LCMV versus MCMV delivered GP33, as prolonged antigen presentation might explain how additional costimulatory molecules are induced. Although a direct measure of antigen levels would be difficult, the level of antigen presentation over time could be measured by transfer of CFSE labeled P14 cells at different times post infection into MCMV-GP33 or LCMV infected mice over time to test the prediction that longer or stronger antigen presentation explains increased induction of costimulatory molecules and redundancy. If possible, such data would add very substantially to the value of the manuscript. At the very least, it is essential to discuss this issue and possibility.

---

## [Author Response]

*1) The part of this study that simply demonstrates contextual variations of costimulatory requirements in viral infections is not novel but conceptually similar to several previous studies, and it would be appropriate to acknowledge them. Thus, for example, the finding that LCMV responses are independent of CD28 (and by inference B7.1, B7.2) for which the original reference (Shahinian et al., Science, 1993) also noted that costimulation requirements are viral context-dependent, or the finding that anti-LCMV but not anti-influenza CD8 T cell responses are unimpaired in the absence of both CD28 and 4-1BBL (DeBenedette et al., JI, 1999), or the epitope repertoire-independence of viral infection contextual costimulation requirements (Lin et al., JI, 2009, and Salek-Ardakani 2011). Thus, the manuscript should be a bit more balanced in acknowledging such previous contributions to the field, particularly when the idea of contextual variation in costimulation requirements has been reviewed recently (Wotzman et al., Immunol Rev 2013)*.

We thank the reviewers for addressing this point and have now more referred to previous work regarding the contextual role for costimulatory molecules. Furthermore, we have discussed this previous work in relation to our novel findings.

*2) The primary limitation of the manuscript is the absence of a comprehensive and systematic framework at least delineating if not explaining the shifting matrix of costimulatory requirements/redundancies in distinct infection/immunisation situations in vivo. Thus, at least for the pair of LCMV (Armstrong) and MCMV, an experiment of the kind shown in*
Figure 7*, but with a more complete set of combinations, and comparing LCMV and MCMV, would be very useful. Such experiments will be best done using transferred p14-TCR-transgenic CD8 T cells to ‘control’ for TCR-repertoire-related differences and to allow the use of IFNIR*^*+/+*^
*and IFNIR*^*-/-*^
*p14 CD8 T cells to enable the inclusion of IFN-I-signaling in the analytical matrix.*

*Technically, too, this most novel aspect of the study is somewhat underdeveloped. For example, the blocking studies in*
Figure 7*, show only one time point for the readout and would be more convincing if more than one time point were examined. The authors administer very large amounts of antibodies without indicating any isotype controls, raising worries about the specificity of the blockade and about possible FcR mediated APC effects.*
Figure 7
*is a critical experiment, but it’s hard to determine how strong the result is – it would be useful to see each mouse shown as a single point and all the mice included, rather than just showing a bar graph*.

To provide a more comprehensive framework for the observed costimulatory redundancy we performed new experiments in which we carefully addressed possible redundant pathways for costimulatory molecules in MCMV and LCMV infection. Importantly, the outcome corroborated our previous findings and additionally strengthened the novel findings regarding costimulatory requirements.

In the LCMV model, we dually blocked OX40L and 4-1BBL in CD70^-/-^ or B7.1/2^-/-^ mice (in addition to CD70^-/-^/B7.1/2^-/-^ mice). Furthermore, we either blocked OX40L or 4-1BBL in CD70/B7.1/2^-/-^ mice. Dual blockade of OX40L and 4-1BBL in B7.1/2^-/-^ mice, and OX40L blockade in CD70/B7.1/2^-/-^ mice showed comparable responses to mice in which all costimulatory pathways were abrogated, indicating that the largest effects on LCMV-specific CD8 T cell expansion are found when both B7.1/2-mediated costimulation and interactions among the TNFR/TNF family members OX40/OX40L are abrogated (shown in Figure 7). In addition, we analysed the LCMV-specific CD8 T cell response at different time points. On day 5.5 post LCMV infection, CD70/B7.1/2^-/-^ and CD70/B7.1/2^-/-^ mice with additional OX40L/4-1BBL block have a lower response in the blood compared to WT mice. Similar to the spleen, on day 7 post-infection these differences are observed in the blood as well. This new data is included in Figure 7. In the antibody blocking experiments, we also added control mice receiving a similar amount of antibody (a rat IgG isotype antibody).

We found that in the absence of CD70- or B7.1/2-mediated costimulation, MCMV-specific CD8 T cell responses were severely hampered, and even more so when both pathways were abrogated (Figure 7). This was also observed for GP33-specific responses using MCMV-IE2-GP33 (Figure 7). It has been shown that blockade of either OX40L or 4-1BBL does not impact the initial MCMV-specific CD8 T cell response (34; 35). We addressed whether dual blockade of OX40L- and 4-1BBL signalling hampered MCMV-specific CD8 T cell expansion, but we did not observe any significant differences compared to mice receiving isotype control antibodies (Figure 7—figure supplement 2). As MCMV-specific CD8 T cell responses were already severely hampered and hardly detectable in CD70/B7.1/2^-/-^ mice, we did not additionally block OX40L and 4-1BBL interactions in CD70/B7.1/2^-/-^ mice during MCMV infection.

To control for possible differences in the TCR repertoire, we transferred P14 cells in WT and costimulation deficient mice and infected these mice with LCMV. Upon transfer of WT P14 cells we found comparable costimulatory requirements as observed for endogenous T cell responses. The largest effects were observed when both CD70- and B7.1/2-mediated costimulation were abrogated (Figure 8). To determine type I IFN related effects on costimulation dependency, IFNAR^-/-^ P14 cells were transferred into WT, CD70^-/-^, B7.1/2^-/-^ and CD70/B7.1/2^-/-^ mice. IFNAR^-/-^ P14 cell expansion was severely hampered in WT mice. In the absence of type I IFN signalling (Figure 8), P14 cells are severely hampered in WT mice and correspondingly also in costimulation deficient mice. We additionally transferred IFNAR^+/+^ and IFNAR^-/-^ P14 cells in WT and costimulation deficient mice that were subsequently infected with MCMV-IE2-GP33, as shown in Figure 8. We found, similar to what is observed for endogenous GP33 responses, that IFNAR^+/+^ P14 cells were highly dependent on costimulatory signals and only slightly hampered when type I IFN signalling was absent. However, GP33-expansion was mostly affected when both type I IFN signalling and costimulatory interactions were abrogated. We feel we have herewith addressed all questions of the referees.

*3) It is necessary to explore mechanistic explanations for the inability of IFN-I (or LCMV-infected mouse day 2 serum) to allow MCMV-GP33-specific P14 CD8 T cells to become* Cd80/86 *independent (*Figure 5*). A GP33-negative LCMV-co-infection (instead of recombinant IFN-I or LCMV-infected mouse serum) might, for example, be useful to test. If there is no reconstitution of* Cd80/86 *independence for MCMV-immunisation even so, then alternative complex explanatory interactions, possibly requiring the inclusion of antigen presentation kinetics and/or infected/presenting APC-specific issues, will need to be considered and addressed substantively*.

*In this context, while the authors show that LCMV infected mice have much higher levels of type I IFN as well as much higher levels of costimulatory molecules than MCMV infected mice, they do not explore the relationships involved any further. Does type I IFN regulate the levels of the costimulatory molecules? For example, does IFNAR blockade lower any of the costimulatory molecules in*
Figure 6*, or is the differential induction in MCMV versus LCMV infection due to direct response to the viral particles? Some efforts to gain insights into the ‘contextual’ role would add value*.

We agree with the reviewers that this is a very interesting aspect to examine. We performed a co-infection experiment using MCMV-Smith and LCMV Armstrong and we found that even upon this co-infection, MCMV-specific CD8 T cells are still dependent on B7.1/2-mediated signals, whereas LCMV-specific T cells are not (Figure 5). The costimulatory requirements seem thus very locally regulated. We addressed possible explanations for the contextual role in the Discussion. To identify the exact mechanism of T cell priming in MCMV and LCMV infection is beyond the scope of this manuscript.

We addressed whether the high levels of type I IFN upon LCMV infection were responsible for the elevated expression levels of costimulatory molecules in the LCMV induced environment. Upon IFNAR blockade we found that on CD11b^+^ cells, the expression of all costimulatory molecules was diminished, however on CD11c^+^ cells only CD86 and 4-1BBL expression was affected. Thus the high levels of type I IFN are only partially responsible for the elevated expression levels of costimulatory ligands that are found. This data is shown in Figure 6—figure supplement 1.

*4) As noted above, data with transferred P14-TCR-transgenic CD8 T cells in all contexts tested would be needed to address the question of whether costimulatory requirement distinctions in LCMV, MCMV, LM and SLP contexts are independent of differences, if any, in the GP33 peptide-specific TCR repertoires*.

We have added data in which TCR transgenic P14 cells are transferred and subsequently primed under different conditions: MCMV-IE2-GP33, LCMV, LM-GP33 (Figure 3). Comparable to the endogenous GP33-specific T cell responses, the expansion of the TCR transgenic P14 cells was dependent on B7.1/2-mediated costimulation upon MCMV-IE2-GP33 and LM-GP33 infection. However, B7.1/2-mediated signals were dispensable during LCMV infection. These data further exclude that differences in the TCR repertoire explain the distinct costimulatory requirements in different pathogenic conditions.

*5) It would be preferable to test all conclusions carefully in situations where the extent of CD8 T cell expansion is similar between the LCMV- versus MCMV-groups being compared, since differences in the degree of expansion may have contributory consequences not apparent in the situation*
Figure 1.

In MCMV-Smith infection, the expansion of M45-specific CD8 T cells is comparable to the expansion of GP33-specific CD8 T cells in LCMV infection, which indicates that the extent of CD8 T cell expansion for dominant epitopes in LCMV and MCMV infection is actually similar. Furthermore, even upon a lower dose of LCMV (2 × 10^2^ PFU/mouse), we still observe that B7.1/2-mediated costimulation is dispensable for LCMV-specific CD8 T cell expansion. To investigate the GP33-specific response in an MCMV context, we have used a MCMV-mutant expressing the GP33 epitope in the IE2 protein. This leads to a lower response due to promoter specific reasons. Indeed, a MCMV-mutant expressing the GP33 epitope in the M45-gene, inducing a GP33-specific response following the normal pattern of expansion and contraction, elicits in the acute phase of infection a higher GP33-specific response than MCMV-IE2-GP33 (Figure 3). Also, in this case we found a high B7.1/2-costimulatory requirement, which further indicates that the degree of expansion is comparable between the LCMV versus MCMV groups, and that possible differences in expansion is not implicated in differences in costimulation redundancy.

*6) It is somewhat surprising, given the importance of littermate controls, that the manuscript uses ‘wild-type’ from a commercial source as a control for in-house knockout mice. It would be useful to have some clarification of this issue*.

We are well aware of possible issues regarding littermate controls and therefore we have bred back the knockout mice onto the background of the B6 mice from the commercial source. We have explained this in the Materials and methods of the revised version.

*7) The data (*Figure 4*) suggest that there are no differences between co-stimulatory requirements for primary versus secondary CD8 T cells. There is some controversy in the extant literature regarding the co-stimulation requirements of ‘memory’ CD8 T cells (*[25]*;*
[12]*). In fact, ‘memory’ CD8 T cells may not respond as well as naive CD8 T cells (*[9]*), and type I interferons may negatively affect memory CD8 T cell proliferation (*[33]*) under certain circumstances. If the present manuscript is to address co-stimulatory requirements of memory CD8 T cells, it would be appropriate to have these issues acknowledged and addressed*.

We agree with the reviewers regarding the literature on this topic. Indeed, it was reported that in vitro memory CD8 T cells do not require costimulatory signals (25). However, we and others have found that depending on the context in vivo costimulatory molecules can be critical for expansion of memory virus-specific CD8 T cells ([12]; Fuse, Zhang and Usherwood, 2008). We have discussed these issues in the Discussion of the revised manuscript.

*8) The title does not reflect the novel aspects of the study; it would be useful for the title to reflect that the new finding is the redundancy of costimulatory pathways for CD8 T cell responses to LCMV, rather than simply the context-dependent nature of costimulation*.

Following the reviewers’ suggestion, we have changed the title of the manuscript to: “The viral context instructs the redundancy of costimulatory pathways in driving CD8 T cell expansion.”

[Editors' note: further revisions were requested prior to acceptance, as described below.]

*A remaining question is: does LCMV induce more costimulation through more inflammation or more antigen? It would be of interest to determine the level of viral antigen in LCMV versus MCMV delivered GP33, as prolonged antigen presentation might explain how additional costimulatory molecules are induced. Although a direct measure of antigen levels would be difficult, the level of antigen presentation over time could be measured by transfer of CFSE labeled P14 cells at different times post infection into MCMV-GP33 or LCMV infected mice over time to test the prediction that longer or stronger antigen presentation explains increased induction of costimulatory molecules and redundancy. If possible, such data would add very substantially to the value of the manuscript. At the very least, it is essential to discuss this issue and possibility*.

We thank the reviewers for addressing this point. It may be indeed interesting to know whether the increased costimulation during LCMV infection is through more inflammation or more antigen. Both MCMV and LCMV are natural mouse pathogens, and both viruses have more extensive replication as compared to vesicular stomatitis virus (VSV) for example (Kundig TM et al., 1996, Immunity). However, whereas LCMV Armstrong is eventually cleared from the body, MCMV becomes latent. It is possible that differences exist between MCMV and LCMV infections with respect to the duration of antigen presentation during the acute phase but virus levels peak in the spleen and liver around day 4 post-infection in case of MCMV (Cicin-Sain L, 2008, J. Virol.; Stacey MA et al., 2011, J.Immunol) but also in case of LCMV (Buchmeier MJ et al., 1980, Adv. Immunol).

We think, however, that the proposed experiment may not enable us to answer this question. Adoptive transfer of P14 cells in LCMV and MCMV infected mice at different time points would tell more about the time points of optimal priming of these cells, but differences in expansion could still be due to differences in the duration of antigen presentation, or due to differences in the local pro-inflammatory environment (or by both in combination). Thus, such adoptive transfers do not discriminate between prolonged antigen presentation and increased inflammation as both can lead to increased costimulatory molecule expression. Also, curtailing the viral replication would change the pro-inflammatory environment.

To examine whether prolonged antigen expression is different between MCMV and LCMV, one would need to perform mass spectrometry to identify the MHC-bound and presented viral epitopes including GP33. However, considering the low number of infected APCs that can be derived from infected mice, this is currently technically not feasible.

Nevertheless, we believe that this is a valid and important point and have discussed this issue in more detail in the revised manuscript.